# KVComm: Enabling Efficient LLM Communication through Selective KV Sharing

**Xiangyu Shi, Marco Chiesa, Gerald Q. Maguire Jr., Dejan Kostić**
KTH Royal Institute of Technology
{xiangyus,mchiesa,maguire,dmk}@kth.se

## Abstract

Large Language Models (LLMs) are increasingly deployed in multi-agent systems, where effective inter-model communication is crucial. Existing communication protocols either rely on natural language, incurring high inference costs and information loss, or on hidden states, which suffer from information concentration bias and inefficiency. To address these limitations, we propose KVComm, a novel communication framework that enables efficient communication between LLMs through selective sharing of KV pairs. KVComm leverages the rich information encoded in the KV pairs while avoiding the pitfalls of hidden states. We introduce a KV layer-wise selection strategy based on attention importance scores with a Gaussian prior to identify the most informative KV pairs for communication. Extensive experiments across diverse tasks and model pairs demonstrate that KVComm achieves comparable performance to the upper-bound method, which directly merges inputs to one model without any communication, while transmitting as few as 30% of layers' KV pairs. Our study highlights the potential of KV pairs as an effective medium for inter-LLM communication, paving the way for scalable and efficient multi-agent systems[1].

## 1 Introduction

Large Language Models (LLMs) have catalyzed a paradigm shift from isolated model capabilities towards collaborative multi-agent systems (Guo et al., 2024; Tran et al., 2025). CAMEL (Li et al., 2023), AutoGen (Wu et al., 2024), and ChatDev (Qian et al., 2023) have demonstrated the potential of LLMs to collaborate effectively in multi-agent systems, achieving impressive results in various tasks. These systems leverage the strengths of individual LLMs and enable them to work together to solve complex problems that are beyond the capabilities of a single model (Yang et al., 2024a).

While multi-agent systems have shown great promise, they also introduce new challenges, particularly in the area of inter-agent communication. Effective communication between LLMs is crucial for the success of multi-agent systems. Explicit communication through natural language has been explored in several works, enabling the models to share information (Du et al., 2023), coordinate their actions (Sun et al., 2025), and make collective decisions (Yang et al., 2024b).

However, natural language communication leads to high inference costs due to the need for multiple decoding steps, and may not fully capture the rich information that needs to be shared between Large Language Models, as information is lost in the sampling process (Pham et al., 2023; Ramesh & Li, 2025) that occurs as each new token is produced. To address this limitation, recent works have explored alternative communication protocols that leverage the internal representations of LLMs. CIPHER (Pham et al., 2023) proposed to use the embedding space as the medium of communication between LLMs. Namely, they pass the weighted average of the token embeddings from one LLM to another, facilitating more efficient information exchange. Rather than using the embedding space, AC (Ramesh & Li, 2025) transmits the intermediate activations, specifically the last token's hidden state. They replace the last token's hidden state of the **receiver's model** ($\mathcal{M}_r$) with that of the **sender's model** ($\mathcal{M}_s$), allowing a more direct transfer of information. While these methods have shown promising results, they still face challenges in terms of communication efficiency and effectiveness. CIPHER (Pham et al., 2023) still requires multiple decoding steps, which can be costly,

---

[1]Our code and data can be found at https://github.com/Zephyroam/KVComm

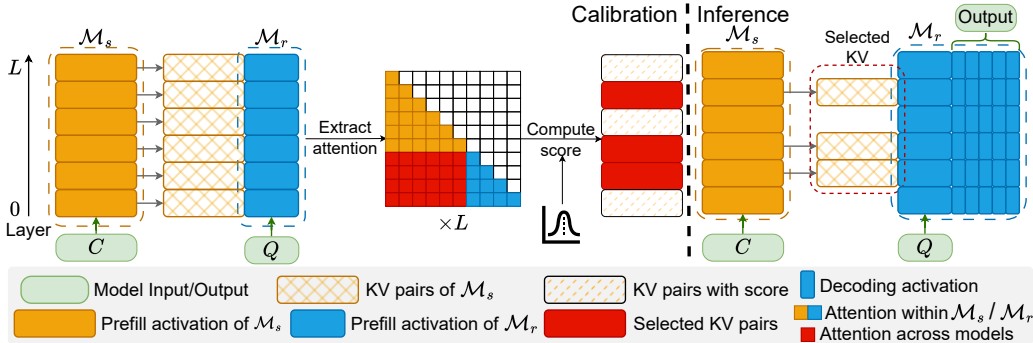

Figure 1: KVComm framework for efficient LLM communication through selective KV sharing.

and AC (Ramesh & Li, 2025) may lead to information loss as only limited activation information is transmitted.

We start with the question: *What is the most effective way to communicate between LLMs?* We argue that an ideal communication protocol should satisfy the following criteria: ① **Effectiveness**: It should enable $\mathcal{M}_r$ to effectively utilize the information from $\mathcal{M}_s$. ② **Efficiency**: It should minimize the computation needed by $\mathcal{M}_s$ and the amount of data transmitted between models. ③ **Generality**: It should be applicable to a wide range of tasks and model architectures, ensuring its versatility in different scenarios. We choose to use activation information as the medium of communication, as no decoding steps are needed for $\mathcal{M}_s$, and $\mathcal{M}_r$ can directly utilize the rich information encoded in the activations. We study different types of activation information (i.e., hidden states and KV pairs), and in Section 2.2, we show that hidden states suffer from information concentration bias, where the last token's hidden state contains most of the information needed for the model's output. This makes it challenging to design an effective communication protocol using the last token's hidden state. Furthermore, we find that using all tokens' hidden states from a single layer of the sender $\mathcal{M}_s$ does not guarantee effective communication. A dilemma arises: if the hidden states are taken from the early layers of $\mathcal{M}_s$, the computation benefit is limited since the computation cost is similar to concatenating the two inputs; if the hidden states are prepended to the later layers of $\mathcal{M}_r$, the performance drops significantly.

Based on these observations, we propose **KVComm**, a novel communication protocol that enables efficient communication between LLMs through selective sharing of KV pairs. KV pairs are the most representative activation information in each layer, and sharing them does not interact with the hidden states of $\mathcal{M}_r$ directly, while $\mathcal{M}_r$ can decide how to utilize the information through the attention mechanism. To further improve the efficiency of communication, we propose a selection strategy to choose which (potentially non-contiguous) layers' KV pairs to share. We formulate hypotheses that (**H1**) *KV pairs from intermediate layers encode transferable semantic knowledge*, and (**H2**) *KV pairs from layers exhibiting stronger attention distributions are more effective for communication*. These hypotheses are validated by our experiments in Sections 4.3 and 4.5. Based on these hypotheses, we define attention importance scores for each layer based on the average attention weights assigned to the context tokens. We also apply a Gaussian distribution centered at a certain layer as a prior on the attention importance scores. The intuition is that the Gaussian distribution encourages selecting layers around a certain depth, which aligns with hypothesis **H1**. The general framework is illustrated in Figure 1.

We evaluate KVComm on a diverse set of tasks with nine model pairs (see Section 4.1), showing that it consistently outperforms existing communication protocols while significantly reducing the data transmitted between models. In summary, our work makes three key contributions:

- We evaluate different types of activation information for communication between LLMs, and identify the limitations of using hidden states as the medium of communication. We show that the last token's hidden state suffers from information concentration bias, and point out a dilemma that arises when using all tokens' hidden states.
- We propose KVComm, a novel communication protocol that enables efficient communication between LLMs through selective sharing of KV pairs. We design a selection strategy based on attention importance scores and a Gaussian prior to choose which layers' KV pairs to share.

This is the first approach that makes it possible to choose non-contiguous layers of KV. Moreover, we show the feasibility of using a single context/question pair for guiding the selection for a given pair of models, prior to deployment.

- We conduct extensive experiments on a diverse set of tasks and model pairs, demonstrating that KVComm enables effective and efficient communication between LLMs, achieving comparable performance to the Skyline method, which is the upper-bound and directly merges the inputs without any communication, while reducing the computation costs by 2.5x to 6x. In particular, KVComm enables up to a 3x reduction in communication relative to approaches that transmit the entire set of KV pairs. Moreover, we demonstrate the performance benefits of non-contiguous selection of KV layers. Finally, we demonstrate the increase in performance that KVComm brings even over Skyline on two datasets, further illustrating the need to communicate in a non-strictly textual manner.

## 2 PROBLEM AND MOTIVATION

### 2.1 PROBLEM FORMULATION

We formally define the problem of solving a contextual task through the communication of two LLMs: $\mathcal{M}_s$ and $\mathcal{M}_r$. $\mathcal{M}_s$ takes as input a context $C$, and generates the required information $I_C$ to be communicated. $\mathcal{M}_r$ takes as input the query $Q$ and the information $I_C$ from $\mathcal{M}_s$, and produces the final output. In this work, we limit the choices of the two LLMs to (1) two instances of the same LLM, and (2) two models that are fine-tuned versions of the same base LLM. The objective is to design a communication protocol that jointly optimizes the communication, computation efficiency, and the information fidelity between $\mathcal{M}_s$ and $\mathcal{M}_r$.

### 2.2 WHY HIDDEN STATES FALL SHORT

When Decoder-Only LLMs infer, the input information flows through the model in the form of activation values, which refer to the intermediate results output by each decoder layer during the forward pass. We refer to the intermediate activation values that are passed between adjacent layers as hidden states. We also consider the KV pairs used in the attention mechanism within each layer as another type of activation information. In this section, we investigate the effectiveness of using hidden states as the medium of communication by studying two questions: *How important are hidden states of tokens at different positions in the sequence?* (Section 2.2.1) *Are hidden states of all tokens effective for communication?* (Section 2.2.2)

#### 2.2.1 TOKEN IMPORTANCE AT DIFFERENT POSITIONS

We begin with a simple experiment examining how token positions affect performance. Using Llama-3.1-8B on MMLU Social Science, we remove or retain the hidden state of **only** specific tokens at a given layer and measure the performance change. As shown in Figure 2, different tokens vary in importance across layers, with the last token becoming most critical in later layers. This aligns with the intuition that the last token is often the most relevant to the current prediction. Thus, the last token's hidden state carries the most influential information for both model output and inter-LLM communication. Results on additional datasets and models are provided in Appendix C.

To ensure efficient communication with hidden states built on this observation, two conditions must hold: (1) $\mathcal{M}_s$ must send at least the last token's hidden state, and (2) the communication protocol should preserve $\mathcal{M}_r$'s last token state as much as possible. The protocol in Ramesh & Li (2025) either replaces $\mathcal{M}_r$'s last token state with that of $\mathcal{M}_s$ or averages the two, but both cause information loss in $\mathcal{M}_r$'s last token state, harming its performance.

#### 2.2.2 UTILIZING ALL TOKENS

Another straightforward approach to ensure the last token's hidden state is preserved is to prepend all tokens' hidden states from $\mathcal{M}_s$ to $\mathcal{M}_r$. The experiments on HotpotQA with Llama-3.1-8B, presented in Figure 3, demonstrate that prepending all tokens' hidden states from $\mathcal{M}_s$ to $\mathcal{M}_r$ is effective if the hidden states are taken from the early layers of $\mathcal{M}_s$ and prepended to the early layers

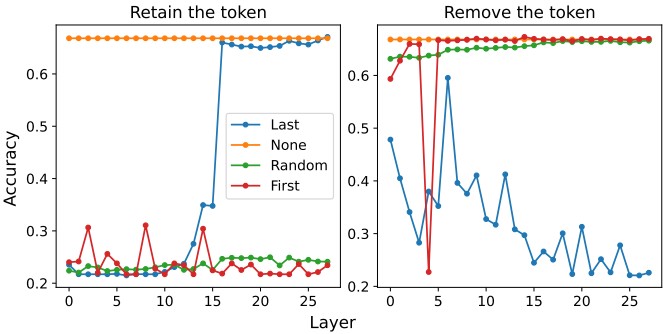 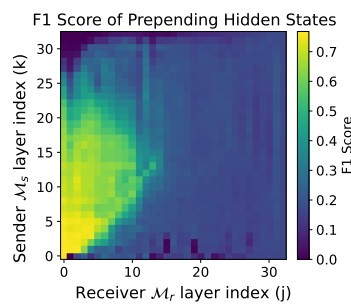

Figure 2: Compared to other token positions, the last token's hidden state is the most critical, especially in later layers.

Figure 3: Prepending helps only when using early-layer hidden states.

of $\mathcal{M}_r$. Appendix D shows experimental results on other datasets. We find that this method is caught in a dilemma: (1) if the hidden states are taken from the early layers of $\mathcal{M}_s$, the computation benefit is limited since it is similar to concatenating the two inputs; (2) if the hidden states are prepended to the later layers of $\mathcal{M}_r$, the performance drops significantly.

These findings suggest that while utilizing all tokens' hidden states can preserve the last token's information, it does not guarantee effective communication between LLMs.

## 3 EFFICIENT LLM COMMUNICATION THROUGH SELECTIVE KV SHARING

We propose a simple yet effective communication protocol that enables efficient communication between LLMs by selectively sharing KV pairs. This approach addresses the limitations observed in previous methods by ensuring that the most critical information is preserved. Our design satisfies the three criteria outlined below: it enhances effectiveness by allowing $\mathcal{M}_r$ to utilize essential context (①), improves efficiency by reducing unnecessary computation and transmission overhead (②), and ensures generality by being applicable across diverse tasks and architectures (③).

### 3.1 COMMUNICATION FRAMEWORK

For a given context $C$ and query $Q$, $\mathcal{M}_s$ processes the context $C$ and runs one forward pass (prefill stage) to generate the KV pairs $\{(\mathbf{k}_s^l, \mathbf{v}_s^l)\}$ at each layer $l$, where $l = 1, 2, \ldots, L$ and $L$ is the total number of layers in $\mathcal{M}_s$. We apply a selection strategy to choose a subset of KV pairs $\{(\mathbf{k}_s^{l_i}, \mathbf{v}_s^{l_i})\}$, where $i = 1, 2, \ldots, M$ and $M$ is the number of selected layers. The selected KV pairs are then transmitted to $\mathcal{M}_r$.

$\mathcal{M}_r$ processes the query $Q$ and incorporates the received KV pairs during its forward passes (prefill and decoding stages). Specifically, at each layer $l$ of $\mathcal{M}_r$, if $l$ corresponds to a selected layer $l_i$[2], the KV pairs from $\mathcal{M}_s$ are integrated into the attention mechanism. We simply concatenate the KV pairs from $\mathcal{M}_s$ with those of $\mathcal{M}_r$: $\mathbf{k}_r^l \leftarrow [\mathbf{k}_s^{l_i}; \mathbf{k}_r^l]$, and $\mathbf{v}_r^l \leftarrow [\mathbf{v}_s^{l_i}; \mathbf{v}_r^l]$. This integration allows $\mathcal{M}_r$ to attend to both its own context and the information provided by $\mathcal{M}_s$. After processing the query $Q$ with the integrated KV pairs, $\mathcal{M}_r$ generates the final output.

### 3.2 KV SELECTION STRATEGIES

The communication protocol critically depends on the selection strategy for choosing which KV pairs to transmit from $\mathcal{M}_s$ to $\mathcal{M}_r$. Not all layers or attention heads contribute equally to encoding task-relevant knowledge. A fundamental question when designing selection strategies is: *Which parts of the KV pairs encode the most relevant knowledge for communication?*

Formally, given the set of candidate KV pairs $\{(\mathbf{k}_s^l, \mathbf{v}_s^l)\}_{l=1}^L$, our goal is to select a subset $\mathcal{S} \subseteq \{1, \ldots, L\}$ such that the receiver's output retains maximal information from the sender, given a

---

[2]The layer indices are 1-to-1 matched between $\mathcal{M}_s$ and $\mathcal{M}_r$ since we only consider the case where the two models are the same or fine-tuned versions of the same base LLM.

constraint on the number of selected layers $|\mathcal{S}| = M$, which is determined by the desired communication efficiency. This can be formulated as the following optimization problem:

$$\max_{\mathcal{S} \subseteq \{1,...,L\}, |\mathcal{S}|=M} f(\mathcal{M}_r(Q, \{(\mathbf{k}_s^l, \mathbf{v}_s^l)\}_{l \in \mathcal{S}})),$$

where $f(\cdot)$ is a performance metric (e.g., accuracy, F1 score), and $\mathcal{M}_r(Q, \{(\mathbf{k}_s^l, \mathbf{v}_s^l)\}_{l \in \mathcal{S}})$ denotes the output of the receiver model given the query $Q$ and the selected KV pairs. Since direct computation of this objective is intractable, we instead propose two hypotheses **H1** and **H2** that serve as priors for designing practical heuristics.

The first hypothesis **H1** is that *KV pairs from intermediate layers contain the most readily transferable semantic knowledge*. Prior analyses (Jawahar et al., 2019; Geva et al., 2020) suggest a hierarchy: early layers capture surface patterns, middle layers encode semantic abstractions, and late layers specialize in task predictions. Thus, intermediate KV pairs should carry the richest generalizable information, making them most effective for communication. Experiment results in Section 4.3 support this hypothesis.

Another hypothesis **H2** is that *KV pairs from layers exhibiting stronger attention distributions are more effective for communication*. We quantify this notion of *attention distribution* using the attention importance score $S_a^l$, defined in Equation (1) below. We deem layer $l_i$ to exhibit stronger attention distribution than $l_j$, if $S_a^{l_i} > S_a^{l_j}$. Intuitively, if a head consistently allocates high attention mass to the given tokens, its KV cache encodes salient contextual relations that are critical for the model's reasoning. Attention concentration thus serves as a proxy for the communication value of a KV subset, suggesting that such heads should be prioritized for selection. This hypothesis is also validated by our experiments in Section 4.5.

Our selection strategy is based on these two hypotheses. We first define attention importance scores for each layer, which are calculated as the average attention weights that have been assigned to the context tokens by all heads in that layer during the prefill stage. We then take a Gaussian distribution centered at a certain layer as a prior to select layers with high attention importance scores. The intuition is that the Gaussian prior encourages selecting layers around a certain depth, which aligns with hypothesis **H1** that intermediate layers are more likely to contain transferable knowledge.

Mathematically, the attention importance score for each layer $l$ is computed as:

$$\hat{S}_a^l = \frac{1}{H|Q|} \sum_{h=1}^{H} \sum_{q=1}^{|Q|} \sum_{c=1}^{|C|} a_{h,q,c}^l, \tag{1}$$

where $H$ is the number of attention heads, $|Q|$ is the number of tokens in the query, $|C|$ is the number of context tokens, and $a_{h,q,c}^l$ is the attention weight assigned by head $h$ at layer $l$ from token $q$ to context token $c$. $\hat{S}_a^l$ is then normalized to the range $[0, 1]$ across all layers to obtain the final attention importance score $S_a^l = \frac{\hat{S}_a^l - \min_{l'} \hat{S}_a^{l'}}{\max_{l'} \hat{S}_a^{l'} - \min_{l'} \hat{S}_a^{l'}}$.

We define a Gaussian prior centered at layer $\mu$ with standard deviation $\sigma$ as $P^l = \exp\left(-\frac{(l-\mu)^2}{2\sigma^2}\right)$. The final selection score for each layer $l$ is computed as a weighted combination of the attention importance score and the Gaussian prior:

$$S^l = \alpha S_a^l + (1 - \alpha) P^l,$$

where $\alpha \in [0, 1]$ is a hyperparameter that balances the two components. We then select the top $M$ layers with the highest selection scores $S^l$ to form the subset $\mathcal{S}$ for communication.

For each model pair and dataset, the top $M$ layers are selected based on the selection scores computed from a calibration set. The selected layers are then fixed and used for all samples in the test set. We found that a calibration set as small as a single sample is sufficient to obtain a robust selection that generalizes well to the entire test set, as shown in the experiments in Appendix H.

### 3.3 COMPLEXITY ANALYSIS

We analyze the computational complexity of our KVComm framework compared to baseline methods. Compared to the NLD (Du et al., 2023) method, our method does not require multiple decoding

steps for $\mathcal{M}_s$, which significantly reduces the computation cost. When the number of tokens generated during debate (Du et al., 2023) is large, the computation margin of our method over NLD is on the order of $O(L(T_s + T_r + |Q|)^2 d)$, where $T_s$ and $T_r$ are the number of tokens generated by $\mathcal{M}_s$ and $\mathcal{M}_r$ in the debate, respectively, and $|Q|$ and $d$ are the number of tokens in the query and the hidden dimension of the model, respectively. Compared to the Skyline (Section 4.1) method, our method also reduces the computation cost, especially when $M$ is small. The computation margin of our method over Skyline is on the order of $O(|C|d(L(2|Q| + T_r) - M(|Q| + T_r)))$, where $|C|$ is the number of tokens in the context.

# 4 EXPERIMENTS

## 4.1 EXPERIMENTAL SETUP

**Datasets** We evaluate KVComm on a diverse set of contextual reasoning tasks. Following Ramesh & Li (2025), we synthetically generate two datasets, Countries, which asks questions about countries based on landmark information, and Tipsheets, which requires investment decisions from financial tips. Examples of these two datasets are shown in Table 3 in Appendix B.1. Moreover, we select six benchmarks, including HotpotQA (Yang et al., 2018), QASPER (Dasigi et al., 2021), MuSiQuest (Trivedi et al., 2022), two subsets of LongBench (Bai et al., 2024)(MultiFieldQA-en and 2WikiMQA), and TMATH (Qi et al., 2025). The last dataset is a mathematical problem-solving dataset that contains hints as context. We use ROUGE-L Recall as the evaluation metric for the last dataset, and F1 score for all other datasets. Statistics are summarized in Table 4 in Appendix B.1.

**Models** We conduct experiments on nine different model pairs, shown in Table 5 in Appendix B.3. The model pairs include two instances of the same LLM and two models that are fine-tuned versions of the same base LLM. These models cover different families, including LLaMA (Dubey et al., 2024), Qwen (Qwen et al., 2024), and Falcon (Almazrouei et al., 2023).

**Compared Methods** We compare KVComm with several representative approaches: **Baseline** (no communication between $\mathcal{M}_r$ and $\mathcal{M}_s$), **Skyline** (concatenating context $C$ and query $Q$ as an upper bound), **Natural Language Debate (NLD)** (Du et al., 2023), **CIPHER** (Pham et al., 2023), and **AC** (Ramesh & Li, 2025). Detailed descriptions for these methods are provided in Appendix B.4. Implementation details are provided in Appendix B.2.

## 4.2 COMMUNICATION RESULTS

Table 1 reports results on three model pairs fine-tuned from the same base LLM. The results on other model pairs are provided in Table 8 in Appendix F, which show similar trends. We observe that KVComm consistently outperforms all baseline communication methods across datasets and model pairs. AC can outperform the Baseline method on some datasets, but they are still significantly worse than KVComm and Skyline, as hidden states of $\mathcal{M}_r$ are corrupted during communication.

NLD and CIPHER can achieve performance close to that of KVComm or Skyline on Countries and Tipsheets datasets, which is because these datasets require only a very small and highly salient amount of information to be transferred. For all other datasets, the sender has access to the entire context but not the question, and natural-language communication cannot reliably extract and transmit the task-relevant subset of information. As a result, NLD and CIPHER perform substantially below KVComm on complex, long-context reasoning tasks. We conduct further experiments in Appendix I to eliminate the influence of hyperparameters.

KVComm can achieve comparable performance to Skyline when selecting 70% of layers' KV pairs for communication, demonstrating the effectiveness of our selection strategy. Even when selecting only 30% of layers' KV pairs, KVComm can still outperform most baseline communication methods on many datasets, showing its potential for efficient communication with minimal overhead.

Note that KVComm can outperform Skyline on some datasets. We attribute this to two factors: (1) $\mathcal{M}_s$ may complement $\mathcal{M}_r$ with stronger capabilities in certain aspects, and (2) selective KV sharing provides a regularization effect, which helps $\mathcal{M}_r$ to focus on the most relevant information and

Table 1: Communication results of different methods. Best results are **bolded**, second best underlined (excluding Baseline and Skyline). We report the results with $\mathcal{M}_r$ for Baseline and Skyline for fairness. **KVComm (0.3/0.5/0.7)** denotes selecting 30%/50%/70% of layers' KV pairs for communication, i.e., $M = \lceil 0.3L \rceil$, $M = \lceil 0.5L \rceil$, $M = \lceil 0.7L \rceil$.

| Method | Countries | Tipsheets | HotpotQA | QASPER | MuSiQuest | MultiField -QA-en | 2WikiM -QA | TMATH |
|---|---|---|---|---|---|---|---|---|
| $\mathcal{M}_s$: **huihui-ai/Llama-3.2-3B-Instruct-abliterated**; $\mathcal{M}_r$: **suayptalha/DeepSeek-R1-Distill-Llama-3B** | | | | | | | | |
| **Baseline** | 0.05 | 0.32 | 0.23 | 0.05 | 0.02 | 0.11 | 0.27 | 0.34 |
| **Skyline** | 0.57 | 0.91 | 0.73 | 0.25 | 0.51 | 0.47 | 0.40 | 0.36 |
| **NLD** | 0.43 | 0.72 | 0.43 | 0.10 | 0.18 | 0.09 | 0.30 | 0.33 |
| **CIPHER** | 0.42 | 0.69 | 0.50 | 0.10 | 0.18 | 0.13 | 0.32 | 0.32 |
| **AC (mean)** | 0.03 | 0.45 | 0.25 | 0.05 | 0.02 | 0.13 | 0.23 | **0.35** |
| **AC (replace)** | 0.00 | 0.49 | 0.05 | 0.01 | 0.01 | 0.12 | 0.03 | 0.34 |
| **AC (sum)** | 0.02 | 0.46 | 0.23 | 0.05 | 0.01 | 0.13 | 0.24 | 0.34 |
| **KVComm (0.3)** | 0.46 | 0.45 | 0.46 | 0.09 | 0.28 | 0.15 | 0.28 | **0.35** |
| **KVComm (0.5)** | **0.57** | **0.81** | 0.57 | 0.27 | 0.32 | **0.51** | 0.36 | **0.35** |
| **KVComm (0.7)** | **0.57** | **0.81** | **0.65** | **0.29** | **0.36** | 0.47 | **0.37** | **0.35** |
| $\mathcal{M}_s$: **Orion-zhen/Qwen2.5-7B-Instruct-Uncensored**; $\mathcal{M}_r$: **bespokelabs/Bespoke-Stratos-7B** | | | | | | | | |
| **Baseline** | 0.01 | 0.36 | 0.13 | 0.05 | 0.03 | 0.08 | 0.09 | 0.35 |
| **Skyline** | 0.51 | 0.97 | 0.53 | 0.10 | 0.25 | 0.40 | 0.09 | 0.35 |
| **NLD** | 0.21 | 0.80 | 0.16 | 0.02 | 0.04 | 0.11 | 0.02 | **0.35** |
| **CIPHER** | 0.04 | 0.60 | 0.03 | 0.01 | 0.03 | 0.07 | 0.03 | 0.34 |
| **AC (mean)** | 0.00 | 0.00 | 0.03 | 0.00 | 0.00 | 0.08 | 0.01 | 0.01 |
| **AC (replace)** | 0.00 | 0.00 | 0.00 | 0.00 | 0.00 | 0.00 | 0.00 | 0.00 |
| **AC (sum)** | 0.00 | 0.00 | 0.02 | 0.00 | 0.00 | 0.07 | 0.04 | 0.03 |
| **KVComm (0.3)** | 0.04 | 0.26 | 0.02 | 0.01 | 0.01 | 0.09 | 0.08 | 0.31 |
| **KVComm (0.5)** | 0.19 | 0.88 | 0.28 | 0.07 | 0.12 | 0.26 | 0.10 | 0.33 |
| **KVComm (0.7)** | **0.41** | **0.89** | **0.41** | **0.21** | **0.25** | **0.29** | **0.15** | 0.34 |
| $\mathcal{M}_s$: **ehristoforu/falcon3-ultraset**; $\mathcal{M}_r$: **huihui-ai/Falcon3-7B-Instruct-abliterated** | | | | | | | | |
| **Baseline** | 0.08 | 0.36 | 0.21 | 0.06 | 0.04 | 0.09 | 0.23 | 0.31 |
| **Skyline** | 0.56 | 0.95 | 0.76 | 0.32 | 0.56 | 0.51 | 0.45 | 0.37 |
| **NLD** | **0.46** | 0.80 | 0.52 | 0.19 | 0.25 | 0.11 | 0.24 | 0.15 |
| **CIPHER** | 0.30 | 0.19 | 0.27 | 0.02 | 0.07 | 0.06 | 0.25 | 0.17 |
| **AC (mean)** | 0.01 | 0.46 | 0.25 | 0.06 | 0.04 | 0.09 | 0.23 | 0.31 |
| **AC (replace)** | 0.00 | 0.49 | 0.12 | 0.00 | 0.01 | 0.13 | 0.17 | 0.31 |
| **AC (sum)** | 0.01 | 0.46 | 0.25 | 0.06 | 0.03 | 0.10 | 0.24 | 0.31 |
| **KVComm (0.3)** | **0.46** | 0.69 | 0.59 | 0.19 | 0.40 | 0.35 | 0.29 | 0.32 |
| **KVComm (0.5)** | 0.40 | 0.92 | **0.63** | 0.25 | **0.44** | 0.45 | **0.34** | 0.35 |
| **KVComm (0.7)** | 0.19 | **0.96** | 0.55 | **0.26** | 0.42 | **0.51** | 0.31 | **0.36** |

avoid wasting its capacity on less important signals. This also explains why using fewer layers can sometimes yield better performance than using more.

Also note that the performance gain of KVComm is not substantial on TMATH. We attribute this to that pretraining gives LLMs solid capabilities in mathematical reasoning, which may not dramatically benefit from additional context or hints. Moreover, AC performs relatively well on this dataset, which we consider is because the hints contain information about questions, so even if the last token's hidden states are corrupted, it can still generate some useful information.

## 4.3 BENEFIT OF SELECTIVE KV OVER ONE CONTIGUOUS CHUNK

DroidSpeak (Liu et al., 2024b) chooses to use one contiguous chunk of context for communication between LLMs. Despite different problem settings, we evaluate KVComm by replacing the selection strategy with two hyperparameters, which are two layer indices $\text{layer}_{\text{from}}$ and $\text{layer}_{\text{to}}$, then all layers between $\text{layer}_{\text{from}}$ and $\text{layer}_{\text{to}}$ are selected for communication. This is equivalent to using one contiguous chunk of context for communication. We vary them to select different chunks of layers.

Figure 4 shows that using a single contiguous chunk for communication yields good performance only in a small region of the hyperparameter space, making it tricky to find the right hyperparameters. In contrast, the scatter and curve plots in Figure 5 demonstrate that KVComm consistently achieves the best or even outperforms the best contiguous chunk setting for the same number of layers. Line plots in Figure 6 show that contiguous chunks are most effective when taken from in-

termediate layers, consistent with hypothesis **H1** in Section 3.2. All results are on HotpotQA with the Llama-3.1-8B pair, with more in Appendix O.

## 4.4 ABLATION STUDY ON SELECTION STRATEGY

Table 2 compares KVComm with random selection. We find that KVComm consistently outperforms random selection across different datasets and selection ratios. When the ratio is high (i.e., 0.7), the performance gap between our selection strategy and random selection becomes smaller, as more layers are selected and the impact of the selection strategy is reduced. However, when the ratio is low (i.e., 0.3), our selection strategy significantly outperforms random selection, demonstrating its effectiveness in selecting the most informative layers for communication. Comparison results on other model pairs are provided in Table 9 in Appendix G, which show similar trends.

Table 2: Comparison with random selection. Best results for each selection ratio are **bolded**.

| Method | Countries | Tipsheets | HotpotQA | QASPER | MuSiQuest | MultiField-QA-en | 2WikiM-QA | TMATH |
|---|---|---|---|---|---|---|---|---|
| $\mathcal{M}_s$: **huihui-ai/Llama-3.2-3B-Instruct-abliterated**; $\mathcal{M}_r$: **suayptalha/DeepSeek-R1-Distill-Llama-3B** | | | | | | | | |
| Random (0.3) | 0.05 | 0.32 | 0.18 | 0.07 | 0.01 | 0.06 | 0.17 | 0.33 |
| **KVComm (0.3)** | **0.46** | **0.45** | **0.46** | **0.09** | **0.28** | **0.15** | **0.28** | **0.35** |
| Random (0.5) | 0.26 | 0.44 | 0.37 | 0.08 | 0.10 | 0.09 | 0.21 | 0.34 |
| **KVComm (0.5)** | **0.57** | **0.81** | **0.57** | **0.27** | **0.32** | **0.51** | **0.36** | **0.35** |
| Random (0.7) | 0.57 | 0.82 | 0.62 | 0.20 | 0.34 | 0.30 | 0.28 | 0.35 |
| **KVComm (0.7)** | **0.57** | 0.81 | **0.65** | **0.29** | **0.36** | **0.47** | **0.37** | **0.35** |

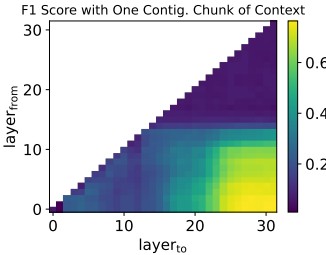

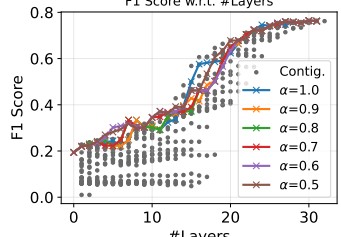

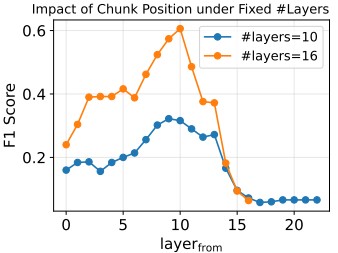

Figure 4: Effective communication with limited hyperparameters.

Figure 5: KVComm achieves nearly the best or even outperforms contig. chunks.

Figure 6: Chunks in intermediate layers achieve the most effective communication.

## 4.5 ATTENTION DISTRIBUTION ANALYSIS

We validate hypothesis **H2** in Section 3.2 by selecting layers with different attention importance scores for communication. We select 9 layers with different levels of attention importance scores, and test the communication performance with Llama-3.2-3B model. The results are shown in Figure 7. We find that selecting layers with higher scores can achieve better performance, while selecting layers with lower scores can diminish the performance. This validates hypothesis **H2** that layers with higher attention importance scores are more effective for communication.

## 4.6 SYSTEM EFFICIENCY

Mathematically, we have shown in Section 3.3 that KVComm can reduce the computation cost compared to Skyline. We validate this through experiments on the Llama-3.2-3B model pair with Tipsheets and MultiFieldQA-en datasets. We report the relative FLOPs of KVComm and Skyline over AC in Figure 8. NLD and CIPHER are not included since they require multiple decoding steps for $\mathcal{M}_s$, which makes the computation cost significantly higher than AC. We find that KVComm has a significant computation advantage over Skyline, especially when selecting fewer layers for communication. This demonstrates the efficiency of our KVComm framework in enabling effective communication with reduced computational overhead by 2.5x to 6x.

In addition to FLOPs, we also report the memory consumption among methods. KVComm similarly shows a substantial memory advantage over Skyline, as the reduced number of communicated layers not only lowers computation but also alleviates memory pressure. On Tipsheets, KVComm uses 23% to 73% less memory than Skyline. This further highlights the efficiency of our framework in achieving lightweight inter-model communication.

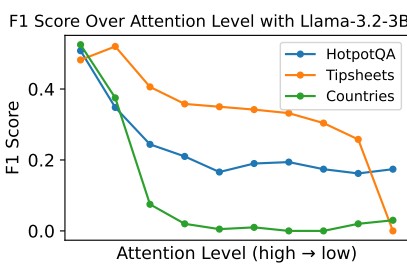

Figure 7: Better communication performance with higher attention level.

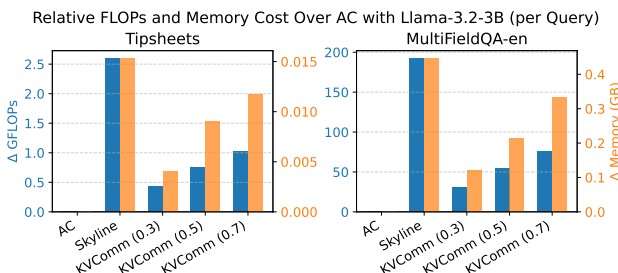

Figure 8: KVComm requires less computation and memory compared to Skyline.

## 5 RELATED WORK

**LLM Inference Acceleration** Lots of work has focused on accelerating LLM inference. Computation-level methods such as FlashAttention (Dao et al., 2022) and Memory-Efficient Attention (Rabe & Staats, 2021) reduce memory and speed up attention; system-level methods such as vLLM (Kwon et al., 2023) and DeepSpeed-Inference (Aminabadi et al., 2022) improve overall throughput and latency; and model-level methods such as quantization (Lin et al., 2024) and pruning (Ma et al., 2023) reduce model size and complexity. These works mainly focus on working with only one model processing a single long input with the aim of minimizing computation cost. These approaches are orthogonal to ours and can be combined with KVComm to further improve efficiency.

Closest to our work are methods that reuse computation across decoding steps or requests. Gao et al. (2024) introduces a hierarchical KV caching system for all requests; Gim et al. (2024) reuses prompt KV caches across queries by decomposing inputs; Liu et al. (2024c) compresses KV caches into compact bitstreams; and Yao et al. (2025) combines multiple chunks' KV caches by selectively recomputing a few tokens. In contrast, our work targets communication across different LLMs, which is more challenging due to parameter differences. Moreover, while prior methods reuse KV caches uniformly across layers, we enable selective sharing of KV caches from different layers, further improving efficiency. We do not compare with these works since they are orthogonal to ours.

DroidSpeak (Liu et al., 2024b) aims to accelerate inference for queries with shared prefixes. It reuses the partial KV cache of these prefixes among different queries. Specifically, it empirically selects a single contiguous chunk of layers and recomputes the rest with large calibration overhead, whereas our strategy flexibly selects non-contiguous layers with low overhead, without needing to recompute the remaining layers. Despite different problem settings, we compare their contiguous-chunk strategy with ours in Section 4.3, showing the advantages of our approach.

Ye et al. (2025) adjusts KV cache for shared content by referencing a pool of cached examples-termed anchors that store observed cache deviations under varying prefixes. Our work goes beyond this related work by: 1) enabling a different type of communication, where the receiver does not have access to the context, 2) making it possible to efficiently and selectively choose layers of KV pairs that will be transmitted, and 3) being able to work effectively across different models that are fine-tuned from one model.

**Inter-LLM Communication** Communication between multiple LLMs has been explored in several recent works. Most works focus on using natural language as the medium of communication. For example, Du et al. (2023) proposed a natural language debate framework where LLMs iteratively critique each other's answers in natural language to improve the final answer. Liang et al. (2023) followed a similar idea but introduced a judge model to manage the debate process.

CIPHER (Pham et al., 2023) proposed using embedding space as the medium of communication. They pass the weighted average of the token embeddings from one LLM to another. Moreover, AC (Ramesh & Li, 2025) proposed to use the last token's hidden state as the medium of communication. They replace the last token's hidden state of the receiver model with that of the sender model. Instead, we propose to use the KV pairs as the medium, which can preserve more information than just using the last token's hidden state. We also propose a more effective selection strategy for choosing which KV pairs to share, which can further improve efficiency.

**KV Cache Optimization**   Several works have explored optimizing KV caches for a single LLM by (1) compressing the KV caches to reduce memory usage (Ge et al., 2023; Liu et al., 2024a) or (2) managing the KV caches (offloading) to improve the inference speed (Lee et al., 2024; Xiong et al., 2024). As our work focuses on layer-wise selection of KV caches for communication between two LLMs, these methods are orthogonal and can be combined with our method.

## 6  DISCUSSION

In this section, we discuss the limitations, clarify the scope of current design choices, and outline promising directions for future research. Additional discussions can be found in Appendix K.

**Heterogeneous Model Architectures**   Our current KVComm framework assumes that both LLMs share the same base architecture, i.e., identical models or fine-tuned versions of the same base LLM. This is because KV pair structures differ substantially across model families, making direct KV exchange undefined. This architecture dependency is a practical limitation but not a fundamental one. Future work could explore learning latent projections, adapters, or other transformation functions to enable KV exchange across heterogeneous architectures.

**Multiple Sender/Receiver Extensions**   While we focus on a single sender-receiver pair in this work, KVComm can be naturally extended to multiple senders and/or receivers. KVComm can integrate information from multiple senders by concatenating KV caches, and multiple receivers can independently select layers based on their own attention patterns. As shown in Appendix J, we mathematically extend our framework to multiple senders, and perform a preliminary experiment with two senders and one receiver, showing that multiple senders can improve performance due to diversified information sources. However, a systematic study of scaling behaviors in larger multi-agent networks remains future work.

**Context-adaptive Online Calibration**   KVComm currently adopts a fixed layer-selection strategy after calibration for simplicity and computational efficiency, while context-adaptive selection is a promising extension. KVComm can naturally support online and dynamic selection. A demonstration and analysis of this mechanism is provided in Appendix L.

**Layer Selection Priors**   Given our goal of keeping the method simple, efficient, and broadly reproducible, we opt for the Gaussian prior. Other alternatives, such as entropy-weighted or data-driven, are promising but introduce significantly higher complexity, e.g., larger calibration sets, training a selector, or risking overfitting to a particular task distribution. Exploring more sophisticated priors is an interesting direction for future work.

## 7  CONCLUSION

In this work, we identified the potential of using KV pairs as an effective medium for communication between two LLMs. We proposed a novel KVComm framework that enables efficient communication by selectively sharing KV pairs between LLM models. We designed a selection strategy based on attention importance scores and a Gaussian prior to select the most relevant layers. Extensive experiments on diverse datasets and model pairs demonstrated that KVComm can achieve comparable or even superior performance to the Skyline upper bound and other methods, while reducing communication costs by up to 3x. We highlight the generalization ability of our selection strategy, which can be effectively calibrated with only a single sample. Our work opens up new possibilities for efficient inter-LLM communication and paves the way for future research in this direction.

ACKNOWLEDGMENTS

This work was supported by the Knut and Alice Wallenberg Foundation through a Wallenberg Scholar Grant to Prof. Dejan Kostić. This work has been partially supported by Vinnova (the Sweden's Innovation Agency), the Swedish Research Council (agreement No. 2021-04212), and KTH Digital Futures. We thank the anonymous reviewers for their insightful comments and constructive suggestions. We also thank Nicolae Filat for providing an early version of the code for CIPHER comparison and for helpful suggestions on the illustrations.

Computations were enabled by the Berzelius resource provided by the Knut and Alice Wallenberg Foundation at the National Supercomputer Centre. We also acknowledge the EuroHPC Joint Undertaking for awarding this project access to the EuroHPC supercomputer LEONARDO, hosted by CINECA (Italy) and the LEONARDO consortium through an EuroHPC Development Access call.

REPRODUCIBILITY STATEMENT

We provide detailed descriptions of the datasets, model pairs, and implementation details in Appendix B. The code and synthetic datasets, Countries and Tipsheets, are uploaded to GitHub to facilitate reproducibility upon the publication of this work.

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

## A  THE USE OF LARGE LANGUAGE MODELS (LLMS)

Large language models, including ChatGPT, were employed to provide assistance in improving the clarity, coherence, and fluency of the manuscript. These tools were used solely for language refinement, and all scientific content and interpretations remain the responsibility of the authors.

## B  EXPERIMENTAL SETUP

In this appendix, we provide more details about the experimental setup, including dataset details, implementation details, fine-tuned model pairs, and descriptions of compared methods.

### B.1  DATASET

We provide sample prompts and expected answers for the Countries and Tipsheets datasets in Table 3, which are inspired by Ramesh & Li (2025). We also provide the statistics of all datasets used in our experiments in Table 4. HotpotQA, QASPER, MuSiQuest, and TMATH datasets are randomly sampled from their original datasets to reduce the evaluation cost. Extended results on the full datasets are provided in Appendix E.

Table 3: Sample prompts and expected answers for Countries and Tipsheets datasets inspired by Ramesh & Li (2025).

| Dataset | Role | Content |
|---|---|---|
| Countries | $C$ | *Uma is at the Mahaffie House.* |
| | $Q$ | *Which country is Uma located in?* |
| | Answer | *United States* |
| Tipsheets | $C$ | *Atlas LLC is under pressure amid softer trends; EPS -17%; won a sizable customer contract but faces a lawsuit. Sable LLC shows clear momentum and improving execution; authorized a buyback but reported a cyber incident. Trace LLC looks balanced with a mixed near-term setup.* |
| | $Q$ | *You must invest in exactly one company from Atlas LLC, Sable LLC, Trace LLC. Which do you choose?* |
| | Answer | *Sable LLC* |

Table 4: Statistics of the datasets in our experiments.

| Dataset | Size |
|---|---|
| Countries | 200 |
| Tipsheets | 500 |
| HotpotQA (Yang et al., 2018) | 500 |
| QASPER (Dasigi et al., 2021) | 500 |
| MuSiQuest (Trivedi et al., 2022) | 500 |
| MultiFieldQA-en (Bai et al., 2024) | 150 |
| 2WikiMQA (Bai et al., 2024) | 200 |
| TMATH (Qi et al., 2025) | 300 |

### B.2  IMPLEMENTATION DETAILS

We implement our KVComm framework based on the Hugging Face Transformers library (Wolf et al., 2020), and models are loaded in bfloat16 precision. We set the hyperparameters of our selection strategy as $\mu = L/2$, and $\sigma = 10$, where $L$ is the total number of layers in the model. For NLD and CIPHER methods, we set the number of debate rounds to 2, and the maximum generation length to 256 in the debate process. For KVComm, $\alpha$ is set to $1$ for Llama family models, and $0.8$ for Qwen and Falcon family models. These values are obtained by validating on a left-out set. All experiments are conducted on a cluster of nodes, each equipped with an Intel®Xeon®Platinum 8358 Processor

@ 2.60 GHz and 4 NVIDIA A100 GPUs with 64 GB memory. We obtain the FLOPs with PyTorch Profiler[3].

## B.3 Model Pairs

We conduct experiments on nine different model pairs, shown in Table 5. The first four pairs consist of the same LLMs, while the last five pairs consist of models that are fine-tuned on the same base LLM.

Table 5: Model pairs in the evaluation. $\mathcal{M}_s$ is the sender model, and $\mathcal{M}_r$ is the receiver model.

| Index | $\mathcal{M}_s$ | $\mathcal{M}_r$ | Note |
|---|---|---|---|
| 1 | meta-llama/Llama-3.1-8B-Instruct | meta-llama/Llama-3.1-8B-Instruct | Same model |
| 2 | meta-llama/Llama-3.2-3B-Instruct | meta-llama/Llama-3.2-3B-Instruct | Same model |
| 3 | Qwen/Qwen2.5-7B-Instruct | Qwen/Qwen2.5-7B-Instruct | Same model |
| 4 | tiiuae/Falcon3-7B-Instruct | tiiuae/Falcon3-7B-Instruct | Same model |
| 5 | yuvraj17/EvolCodeLlama-3.1-8B-Instruct | Team-ACE/ToolACE-2-Llama-3.1-8B | Fine-tuned on 1 |
| 6 | huihui-ai/Llama-3.2-3B-Instruct-abliterated | suayptalha/DeepSeek-R1-Distill-Llama-3B | Fine-tuned on 2 |
| 7 | Orion-zhen/Qwen2.5-7B-Instruct-Uncensored | bespokelabs/Bespoke-Stratos-7B | Fine-tuned on 3 |
| 8 | ehristoforu/falcon3-ultraset | huihui-ai/Falcon3-7B-Instruct-abliterated | Fine-tuned on 4 |
| 9 | arcee-ai/Llama-3.1-SuperNova-Lite | deepseek-ai/DeepSeek-R1-Distill-Llama-8B | Fine-tuned on 1 |

## B.4 Compared Method Descriptions

We compare our proposed KVComm framework with the following methods:

- **Baseline**: $\mathcal{M}_r$ processes the query $Q$ *without* any communication from $\mathcal{M}_s$.
- **Skyline**: $\mathcal{M}_r$ directly processes the concatenation of the context $C$ and query $Q$. This serves as an upper bound for performance.
- **Natural Language Debate (NLD)** (Du et al., 2023): Each model generates an initial answer, and then they iteratively critique each other's answers in natural language for a fixed number of rounds. Finally, one model produces the final answer based on the entire debate history. Compared to the original debate style setting, we use an information-transfer style, which explicitly prompts $\mathcal{M}_s$ that it has to summarize the context $C$ in its initial answer. We set the number of debate rounds to 2.
- **CIPHER** (Pham et al., 2023): Similar to NLD, but instead of communicating in natural language, the models communicate by passing the weighted average of the token embeddings from one LLM to another. We use the same prompt as NLD, and set the number of debate rounds to 2.
- **AC** (Ramesh & Li, 2025): Communicate with the last token's hidden state. Replace the last token's hidden state of $\mathcal{M}_r$ with that of $\mathcal{M}_s$. We also test with mean and sum operations.

## C Token Importance at Different Positions

We add more details and experiments related to Section 2.2.1 in this appendix.

### C.1 Detailed Experiment Procedure

We provide a detailed description of the experiment procedure in Section 2.2.1. Considering a model $\mathcal{M}$ with $L$ layers, given an input $X$ with $N$ tokens, we run a partial forward pass until layer $l$ to obtain the hidden states $\{\mathbf{h}_i^l\}_{i=1}^N$. Then, given a specific token position $k$, if we perform the **Retain** operation, we create a modified set of hidden states $\{\tilde{\mathbf{h}}_i^l\}_{i=1}^N$ as follows:

$$\tilde{\mathbf{h}}_i^l = \begin{cases} \mathbf{h}_i^l, & \text{if } i = k \\ \mathbf{0}, & \text{otherwise} \end{cases}$$

---

[3]https://docs.pytorch.org/docs/stable/profiler.html

If we perform the **Remove** operation, we create the modified set of hidden states $\{\tilde{\mathbf{h}}_i^l\}_{i=1}^N$ as follows:

$$\tilde{\mathbf{h}}_i^l = \begin{cases} \mathbf{0}, & \text{if } i = k \\ \mathbf{h}_i^l, & \text{otherwise} \end{cases}$$

We then continue the forward pass from layer $l + 1$ to layer $L$ using the modified hidden states $\{\tilde{\mathbf{h}}_i^l\}_{i=1}^N$ as input, and obtain the final output of the model. We evaluate the model's performance on the task with different token positions $k$ and layer $l$.

## C.2  MORE EXPERIMENTS ON TOKEN IMPORTANCE

We conduct the same experiment as in Section 2.2.1 on other datasets and models to investigate the effect of tokens at different positions in the sequence on the model's output. We report the results on MMLU Social Science, MMLU STEM, and MMLU Humanities using Llama-3.1-8B and Llama-3.2-3B models in Figure 9. We can see that the last token's hidden state plays the most critical role in the latter layers, which is consistent with the observation in Section 2.2.1.

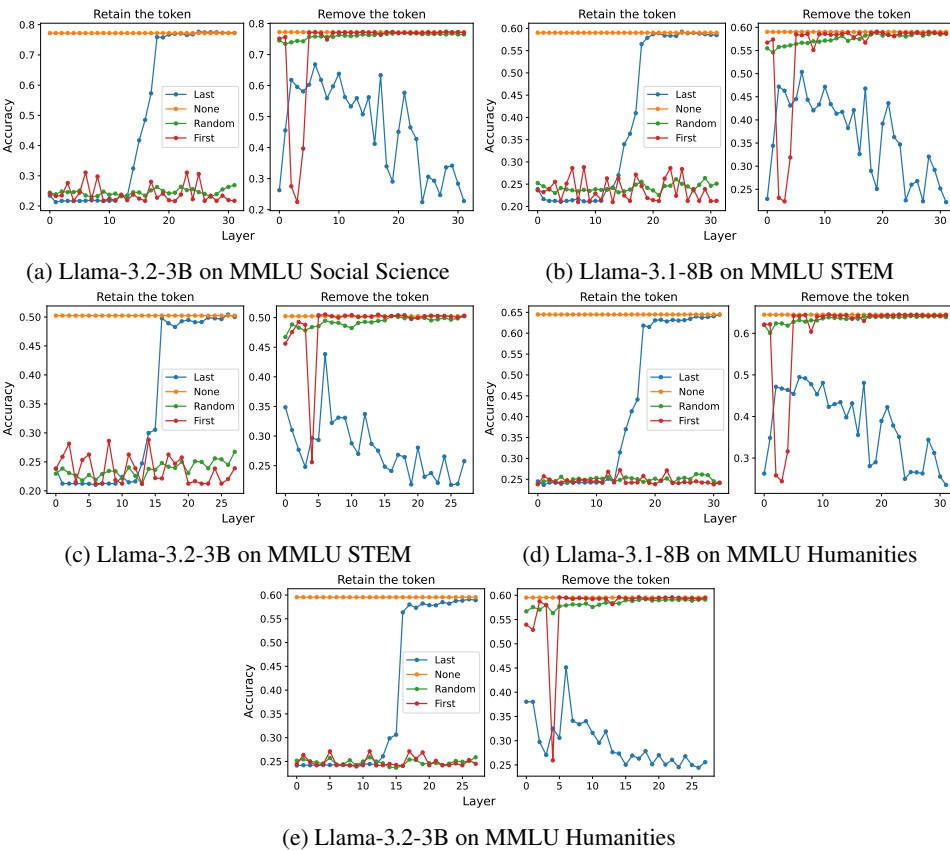

Figure 9: Effect of removing or retaining a token's hidden state across different positions on MMLU Social Science, MMLU STEM, and MMLU Humanities accuracy using Llama-3.1-8B and Llama-3.2-3B models.

# D  UTILIZING ALL TOKENS

We add more details and experiments related to Section 2.2.2 in this appendix.

## D.1  DETAILED EXPERIMENT PROCEDURE

We provide a detailed description of the experiment procedure in Section 2.2.2. Considering two models $\mathcal{M}_s$ and $\mathcal{M}_r$, each with $L$ layers, given $C$ and $Q$ as input, we run a partial forward pass

of $\mathcal{M}_s$ until layer $k$ to obtain the hidden state $\mathbf{H}_k^s \in \mathbb{R}^{|C| \times d}$ for all tokens in $C$, where $|C|$ is the number of tokens in $C$, and $d$ is the hidden dimension. Another partial forward pass of $\mathcal{M}_r$ is run until layer $j$ to obtain the hidden state $\mathbf{H}_j^r \in \mathbb{R}^{|Q| \times d}$ for all tokens in $Q$, where $|Q|$ is the number of tokens in $Q$. We then modify the hidden states of $\mathcal{M}_r$ at layer $j$ by prepending the hidden states from $\mathcal{M}_s$ at layer $k$ as follows:

$$\tilde{\mathbf{H}}_j^r = \begin{bmatrix} \mathbf{H}_k^s \\ \mathbf{H}_j^r \end{bmatrix}$$

We continue the forward pass from layer $j+1$ to layer $L$ using the modified hidden states $\tilde{\mathbf{H}}_j^r$ as input, and obtain the final output of the model. We evaluate the model's performance on the task with different layers $k$ and $j$.

### D.2    MORE EXPERIMENTS ON UTILIZING ALL TOKENS

We conduct the same experiment as in Section 2.2.2 on Countries, Tipsheets, and HotpotQA datasets using Llama-3.1-8B, Llama-3.2-3B, and Qwen2.5-7B models. The results are shown in Figure 10. We can see the results are consistent with the observation in Section 2.2.2.

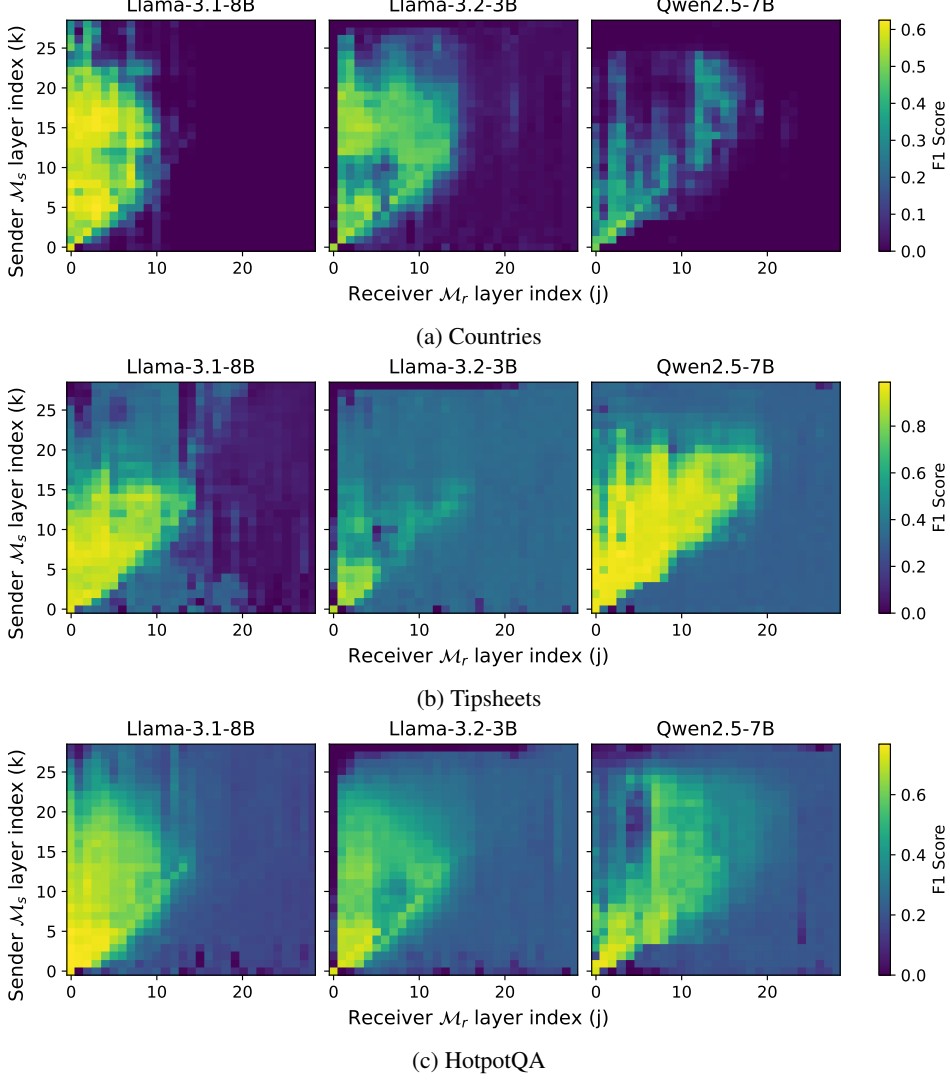

(a) Countries

(b) Tipsheets

(c) HotpotQA

Figure 10: Performance heatmap of prepending the hidden states from certain layers of $\mathcal{M}_s$ to certain layers of $\mathcal{M}_r$ on Countries, Tipsheets, and HotpotQA.

## E EXPERIMENT RESULTS WITH EXTENDED DATASETS

To further validate the effectiveness of our KVComm framework, we conduct experiments on the full datasets of HotpotQA, QASPER, MuSiQuest, mainly HotpotQA-E, QASPER-E, and MuSiQuest-E. Moreover, we include a new human-created summarization dataset, SAMSum, which represents a different task type. The statistics of these datasets are shown in Table 7. We report the results on these extended datasets in Table 6. The results show similar trends as in Section 4.2, demonstrating the robustness of our KVComm framework across different datasets and tasks.

Table 6: Communication results on extended communication tasks. The best results in each block are in **bold**, and the second best results are underlined.

| Method | HotpotQA-E | QASPER-E | MuSiQuest-E | SAMSum |
|---|---|---|---|---|
| $\mathcal{M}_s$: **huihui-ai/Llama-3.2-3B-Instruct-abliterated;** $\mathcal{M}_r$: **suayptalha/DeepSeek-R1-Distill-Llama-3B** | | | | |
| Baseline | 0.22 | 0.03 | 0.06 | 0.26 |
| Skyline | 0.77 | 0.52 | 0.25 | 0.33 |
| NLD | 0.45 | 0.10 | 0.18 | 0.28 |
| CIPHER | 0.51 | 0.10 | 0.20 | 0.28 |
| AC (mean) | 0.24 | 0.03 | 0.06 | 0.26 |
| AC (replace) | 0.06 | 0.00 | 0.01 | 0.26 |
| AC (sum) | 0.23 | 0.03 | 0.06 | 0.26 |
| KVComm (0.3) | 0.44 | 0.25 | 0.11 | 0.25 |
| KVComm (0.5) | 0.61 | 0.36 | 0.25 | 0.28 |
| KVComm (0.7) | **0.71** | **0.38** | **0.30** | **0.29** |
| $\mathcal{M}_s$: **Orion-zhen/Qwen2.5-7B-Instruct-Uncensored;** $\mathcal{M}_r$: **bespokelabs/Bespoke-Stratos-7B** | | | | |
| Baseline | 0.15 | 0.04 | 0.06 | 0.25 |
| Skyline | 0.58 | 0.27 | 0.10 | 0.35 |
| NLD | 0.24 | 0.02 | 0.07 | 0.28 |
| CIPHER | 0.04 | 0.01 | 0.02 | **0.37** |
| AC (mean) | 0.03 | 0.00 | 0.00 | 0.01 |
| AC (replace) | 0.00 | 0.00 | 0.00 | 0.00 |
| AC (sum) | 0.03 | 0.00 | 0.00 | 0.04 |
| KVComm (0.3) | 0.02 | 0.00 | 0.01 | 0.18 |
| KVComm (0.5) | 0.21 | 0.14 | 0.08 | 0.30 |
| KVComm (0.7) | **0.40** | **0.34** | **0.21** | 0.35 |
| $\mathcal{M}_s$: **Orion-ehristoforu/falcon3-ultraset;** $\mathcal{M}_r$: **huihui-ai/Falcon3-7B-Instruct-abliterated** | | | | |
| Baseline | 0.21 | 0.06 | 0.06 | 0.27 |
| Skyline | 0.78 | 0.60 | 0.33 | 0.36 |
| NLD | 0.52 | 0.10 | 0.28 | 0.28 |
| CIPHER | 0.28 | 0.03 | 0.09 | 0.17 |
| AC (mean) | 0.24 | 0.06 | 0.07 | 0.26 |
| AC (replace) | 0.12 | 0.02 | 0.01 | 0.26 |
| AC (sum) | 0.23 | 0.05 | 0.06 | 0.26 |
| KVComm (0.3) | **0.59** | 0.15 | 0.40 | 0.28 |
| KVComm (0.5) | **0.59** | 0.22 | **0.46** | 0.31 |
| KVComm (0.7) | **0.59** | **0.26** | 0.36 | **0.32** |

Table 7: Statistics of extended datasets.

| Dataset | Size |
|---|---|
| HotpotQA-E (Yang et al., 2018) | 7,405 |
| QASPER-E (Dasigi et al., 2021) | 1,726 |
| MuSiQuest-E (Trivedi et al., 2022) | 2,417 |
| SAMSum (Gliwa et al., 2019) | 819 |

## F MORE COMMUNICATION RESULTS

We provide more communication results on different model pairs in Table 8, which show similar trends as in Section 4.2.

Table 8: More communication results of different methods. Best results are **bolded**, second best underlined (excluding Baseline and Skyline). We report $\mathcal{M}_r$ for Baseline and Skyline for fairness. **KVComm (0.3/0.5/0.7)** denotes selecting 30%/50%/70% of layers' KV pairs for communication, i.e., $M = \lceil 0.3L \rceil$, $M = \lceil 0.5L \rceil$, $M = \lceil 0.7L \rceil$.

| Method | Countries | Tipsheets | HotpotQA | QASPER | MuSiQuest | MultiField -QA-en | 2WikiM -QA | TMATH |
|---|---|---|---|---|---|---|---|---|
| $\mathcal{M}_s$: meta-llama/Llama-3.1-8B-Instruct; $\mathcal{M}_r$: meta-llama/Llama-3.1-8B-Instruct | | | | | | | | |
| **Baseline** | 0.00 | 0.05 | 0.19 | 0.02 | 0.01 | 0.07 | 0.06 | 0.35 |
| **Skyline** | 0.62 | 0.92 | 0.74 | 0.35 | 0.54 | 0.56 | 0.52 | 0.36 |
| **NLD** | 0.58 | 0.87 | 0.52 | 0.13 | 0.25 | 0.17 | 0.10 | 0.36 |
| **CIPHER** | 0.57 | 0.84 | 0.57 | 0.13 | 0.25 | 0.15 | 0.10 | 0.36 |
| **AC (mean)** | 0.00 | 0.12 | 0.19 | 0.02 | 0.01 | 0.08 | 0.03 | 0.35 |
| **AC (replace)** | 0.00 | 0.36 | 0.15 | 0.02 | 0.01 | 0.07 | 0.05 | 0.35 |
| **AC (sum)** | 0.00 | 0.09 | 0.20 | 0.02 | 0.01 | 0.09 | 0.04 | 0.35 |
| **KVComm (0.3)** | 0.51 | 0.93 | 0.33 | 0.07 | 0.11 | 0.21 | 0.29 | 0.37 |
| **KVComm (0.5)** | **0.62** | 0.95 | 0.60 | **0.29** | 0.34 | 0.50 | 0.37 | 0.37 |
| **KVComm (0.7)** | **0.62** | **0.96** | **0.69** | **0.29** | **0.39** | **0.53** | **0.38** | **0.38** |
| $\mathcal{M}_s$: meta-llama/Llama-3.2-3B-Instruct; $\mathcal{M}_r$: meta-llama/Llama-3.2-3B-Instruct | | | | | | | | |
| **Baseline** | 0.02 | 0.01 | 0.16 | 0.00 | 0.02 | 0.10 | 0.09 | 0.35 |
| **Skyline** | 0.56 | 0.87 | 0.72 | 0.23 | 0.45 | 0.45 | 0.37 | 0.38 |
| **NLD** | 0.51 | 0.71 | 0.49 | 0.09 | 0.18 | 0.11 | 0.07 | 0.34 |
| **CIPHER** | 0.45 | 0.73 | 0.46 | 0.08 | 0.17 | 0.09 | 0.07 | 0.33 |
| **AC (mean)** | 0.00 | 0.07 | 0.18 | 0.01 | 0.02 | 0.09 | 0.06 | 0.35 |
| **AC (replace)** | 0.01 | 0.37 | 0.13 | 0.01 | 0.02 | 0.06 | 0.03 | 0.34 |
| **AC (sum)** | 0.00 | 0.34 | 0.20 | 0.02 | 0.02 | 0.10 | 0.07 | 0.34 |
| **KVComm (0.3)** | 0.51 | 0.48 | 0.47 | 0.10 | 0.20 | 0.17 | 0.28 | 0.35 |
| **KVComm (0.5)** | 0.55 | 0.79 | 0.58 | 0.24 | 0.27 | 0.47 | **0.35** | 0.36 |
| **KVComm (0.7)** | **0.57** | **0.80** | **0.65** | **0.27** | **0.29** | **0.48** | 0.31 | **0.37** |
| $\mathcal{M}_s$: Qwen/Qwen2.5-7B-Instruct; $\mathcal{M}_r$: Qwen/Qwen2.5-7B-Instruct | | | | | | | | |
| **Baseline** | 0.00 | 0.32 | 0.19 | 0.05 | 0.03 | 0.06 | 0.17 | 0.32 |
| **Skyline** | 0.54 | 0.97 | 0.68 | 0.30 | 0.48 | 0.49 | 0.45 | 0.33 |
| **NLD** | 0.18 | 0.86 | 0.37 | 0.09 | 0.11 | 0.11 | 0.19 | 0.30 |
| **CIPHER** | 0.18 | 0.87 | 0.34 | 0.07 | 0.10 | 0.11 | 0.16 | 0.31 |
| **AC (mean)** | 0.00 | 0.37 | 0.15 | 0.01 | 0.02 | 0.10 | 0.20 | **0.33** |
| **AC (replace)** | 0.00 | 0.35 | 0.02 | 0.00 | 0.00 | 0.10 | 0.09 | 0.32 |
| **AC (sum)** | 0.00 | 0.41 | 0.14 | 0.02 | 0.02 | 0.08 | 0.17 | 0.32 |
| **KVComm (0.3)** | 0.04 | 0.31 | 0.06 | 0.02 | 0.01 | 0.19 | 0.19 | 0.32 |
| **KVComm (0.5)** | **0.57** | 0.92 | 0.49 | 0.18 | 0.20 | 0.40 | 0.25 | 0.32 |
| **KVComm (0.7)** | 0.56 | **0.98** | **0.72** | **0.29** | **0.48** | **0.45** | **0.35** | **0.33** |
| $\mathcal{M}_s$: tiiuae/Falcon3-7B-Instruct; $\mathcal{M}_r$: tiiuae/Falcon3-7B-Instruct | | | | | | | | |
| **Baseline** | 0.06 | 0.33 | 0.19 | 0.04 | 0.04 | 0.09 | 0.21 | 0.31 |
| **Skyline** | 0.57 | 0.95 | 0.70 | 0.24 | 0.50 | 0.49 | 0.48 | 0.35 |
| **NLD** | 0.38 | 0.71 | 0.44 | 0.07 | 0.19 | 0.13 | 0.24 | 0.20 |
| **CIPHER** | **0.47** | 0.63 | 0.41 | 0.03 | 0.19 | 0.09 | 0.21 | 0.21 |
| **AC (mean)** | 0.03 | 0.51 | 0.22 | 0.04 | 0.04 | 0.09 | 0.22 | **0.32** |
| **AC (replace)** | 0.00 | 0.57 | 0.09 | 0.00 | 0.02 | 0.12 | 0.14 | 0.31 |
| **AC (sum)** | 0.04 | 0.51 | 0.22 | 0.04 | 0.03 | 0.09 | 0.22 | **0.32** |
| **KVComm (0.3)** | 0.06 | 0.67 | 0.41 | 0.12 | 0.22 | 0.41 | 0.23 | **0.32** |
| **KVComm (0.5)** | 0.16 | 0.94 | 0.52 | **0.22** | **0.33** | **0.47** | **0.33** | **0.32** |
| **KVComm (0.7)** | 0.23 | **0.96** | **0.54** | **0.22** | 0.32 | **0.47** | 0.29 | **0.32** |
| $\mathcal{M}_s$: yuvraj17/EvolCodeLlama-3.1-8B-Instruct; $\mathcal{M}_r$: Team-ACE/ToolACE-2-Llama-3.1-8B | | | | | | | | |
| **Baseline** | 0.00 | 0.07 | 0.04 | 0.00 | 0.01 | 0.08 | 0.01 | 0.34 |
| **Skyline** | 0.24 | 0.95 | 0.37 | 0.17 | 0.15 | 0.51 | 0.25 | 0.39 |
| **NLD** | 0.29 | 0.82 | 0.17 | 0.04 | 0.05 | 0.13 | 0.02 | 0.34 |
| **CIPHER** | 0.21 | 0.86 | 0.19 | 0.03 | 0.06 | 0.15 | 0.03 | 0.33 |
| **AC (mean)** | 0.00 | 0.31 | 0.03 | 0.00 | 0.01 | 0.11 | 0.01 | 0.34 |
| **AC (replace)** | 0.00 | 0.30 | 0.05 | 0.00 | 0.01 | 0.10 | 0.02 | 0.33 |
| **AC (sum)** | 0.00 | 0.27 | 0.04 | 0.00 | 0.01 | 0.09 | 0.01 | 0.34 |
| **KVComm (0.3)** | 0.12 | 0.95 | 0.12 | 0.05 | 0.04 | 0.26 | 0.19 | 0.36 |
| **KVComm (0.5)** | **0.55** | **0.98** | 0.38 | 0.15 | 0.14 | 0.43 | 0.28 | **0.38** |
| **KVComm (0.7)** | 0.53 | 0.97 | **0.51** | **0.22** | **0.25** | **0.49** | **0.33** | **0.38** |
| $\mathcal{M}_s$: arcee-ai/Llama-3.1-SuperNova-Lite; $\mathcal{M}_r$: deepseek-ai/DeepSeek-R1-Distill-Llama-8B | | | | | | | | |
| **Baseline** | 0.07 | 0.30 | 0.11 | 0.01 | 0.03 | 0.09 | 0.16 | 0.23 |
| **Skyline** | 0.55 | 0.80 | 0.52 | 0.17 | 0.40 | 0.41 | 0.16 | 0.29 |

Continued on next page

Table 8 – continued from previous page

| Method | Countries | Tipsheets | HotpotQA | QASPER | MuSiQuest | MultiField -QA-en | 2WikiM -QA | TMATH |
|---|---|---|---|---|---|---|---|---|
| **NLD** | 0.30 | 0.39 | 0.20 | 0.02 | 0.06 | 0.08 | **0.19** | 0.22 |
| **CIPHER** | 0.47 | 0.71 | 0.27 | 0.03 | 0.11 | 0.14 | 0.14 | 0.18 |
| **AC (mean)** | 0.00 | 0.31 | 0.08 | 0.02 | 0.02 | 0.09 | 0.14 | 0.25 |
| **AC (replace)** | 0.00 | 0.39 | 0.04 | 0.01 | 0.00 | 0.16 | 0.16 | 0.28 |
| **AC (sum)** | 0.02 | 0.34 | 0.07 | 0.02 | 0.02 | 0.08 | 0.16 | 0.24 |
| **KVComm (0.3)** | 0.09 | 0.52 | 0.10 | 0.01 | 0.03 | 0.09 | 0.08 | 0.28 |
| **KVComm (0.5)** | 0.41 | **0.76** | 0.33 | 0.05 | 0.21 | 0.23 | 0.09 | **0.29** |
| **KVComm (0.7)** | **0.53** | **0.76** | **0.47** | **0.12** | **0.28** | **0.31** | 0.14 | **0.29** |

# G  ABLATION STUDY ON SELECTION STRATEGY

We conduct more ablation studies on the selection strategy by comparing with random selection and selection based on only attention importance scores. The results are shown in Table 9, which show similar trends as in Section 4.4.

Table 9: More comparison results with random selection. Best results for each selection ratio are **bolded**.

| Method | Countries | Tipsheets | HotpotQA | QASPER | MuSiQuest | MultiField -QA-en | 2WikiM -QA | TMATH |
|---|---|---|---|---|---|---|---|---|
| $\mathcal{M}_s$: meta-llama/Llama-3.1-8B-Instruct; $\mathcal{M}_r$: meta-llama/Llama-3.1-8B-Instruct | | | | | | | | |
| **Random (0.3)** | 0.02 | 0.35 | 0.24 | **0.07** | 0.04 | 0.07 | 0.12 | 0.35 |
| **KVComm (0.3)** | **0.51** | **0.93** | **0.33** | **0.07** | **0.11** | **0.21** | **0.29** | **0.37** |
| **Random (0.5)** | 0.49 | 0.76 | 0.58 | 0.15 | 0.29 | 0.29 | 0.27 | 0.36 |
| **KVComm (0.5)** | **0.62** | **0.95** | **0.60** | **0.29** | **0.34** | **0.50** | **0.37** | **0.37** |
| **Random (0.7)** | **0.63** | 0.88 | **0.76** | **0.32** | **0.49** | 0.52 | 0.34 | 0.37 |
| **KVComm (0.7)** | 0.62 | **0.96** | 0.69 | 0.29 | 0.39 | **0.53** | **0.38** | **0.38** |
| $\mathcal{M}_s$: Orion-zhen/Qwen2.5-7B-Instruct-Uncensored; $\mathcal{M}_r$: bespokelabs/Bespoke-Stratos-7B | | | | | | | | |
| **Random (0.3)** | 0.00 | 0.09 | 0.00 | 0.00 | 0.00 | 0.06 | 0.01 | **0.31** |
| **KVComm (0.3)** | **0.04** | **0.26** | **0.02** | **0.01** | **0.01** | **0.09** | **0.08** | **0.31** |
| **Random (0.5)** | 0.12 | 0.32 | 0.06 | 0.00 | 0.03 | 0.15 | 0.04 | **0.33** |
| **KVComm (0.5)** | **0.19** | **0.88** | **0.28** | **0.07** | **0.12** | **0.26** | **0.10** | **0.33** |
| **Random (0.7)** | 0.16 | 0.76 | 0.14 | 0.03 | 0.02 | 0.20 | 0.04 | **0.34** |
| **KVComm (0.7)** | **0.41** | **0.89** | **0.41** | **0.21** | **0.25** | **0.29** | **0.15** | **0.34** |
| $\mathcal{M}_s$: ehristoforu/falcon3-ultraset; $\mathcal{M}_r$: huihui-ai/Falcon3-7B-Instruct-abliterated | | | | | | | | |
| **Random (0.3)** | 0.35 | 0.36 | 0.23 | 0.06 | 0.07 | 0.14 | 0.24 | **0.31** |
| **KVComm (0.3)** | **0.46** | **0.69** | **0.59** | **0.19** | **0.40** | **0.35** | **0.29** | **0.32** |
| **Random (0.5)** | 0.23 | 0.42 | 0.27 | 0.09 | 0.08 | 0.15 | 0.28 | 0.31 |
| **KVComm (0.5)** | **0.40** | **0.92** | **0.63** | **0.25** | **0.44** | **0.45** | **0.34** | **0.35** |
| **Random (0.7)** | 0.18 | 0.94 | 0.51 | 0.23 | 0.35 | 0.47 | 0.30 | 0.34 |
| **KVComm (0.7)** | **0.19** | **0.96** | **0.55** | **0.26** | **0.42** | **0.51** | **0.31** | **0.36** |
| $\mathcal{M}_s$: meta-llama/Llama-3.2-3B-Instruct; $\mathcal{M}_r$: meta-llama/Llama-3.2-3B-Instruct | | | | | | | | |
| **Random (0.3)** | 0.02 | 0.29 | 0.11 | 0.06 | 0.02 | 0.07 | 0.16 | 0.34 |
| **KVComm (0.3)** | **0.51** | **0.48** | **0.47** | **0.10** | **0.20** | **0.17** | **0.28** | **0.35** |
| **Random (0.5)** | 0.28 | 0.44 | 0.30 | 0.06 | 0.06 | 0.06 | 0.19 | 0.35 |
| **KVComm (0.5)** | **0.55** | **0.79** | **0.58** | **0.24** | **0.27** | **0.47** | **0.35** | **0.36** |
| **Random (0.7)** | 0.54 | **0.81** | 0.62 | 0.21 | **0.30** | 0.30 | 0.26 | 0.36 |
| **KVComm (0.7)** | **0.57** | 0.80 | **0.65** | **0.27** | 0.29 | **0.48** | **0.31** | **0.37** |
| $\mathcal{M}_s$: Qwen/Qwen2.5-7B-Instruct; $\mathcal{M}_r$: Qwen/Qwen2.5-7B-Instruct | | | | | | | | |
| **Random (0.3)** | 0.00 | 0.34 | 0.05 | 0.00 | 0.00 | 0.08 | 0.10 | **0.30** |
| **KVComm (0.3)** | **0.04** | **0.31** | **0.06** | **0.02** | **0.01** | **0.19** | **0.19** | **0.32** |
| **Random (0.5)** | 0.00 | 0.32 | 0.10 | 0.02 | 0.02 | 0.10 | 0.16 | **0.32** |
| **KVComm (0.5)** | **0.57** | **0.92** | **0.49** | **0.18** | **0.20** | **0.40** | **0.25** | **0.32** |
| **Random (0.7)** | 0.41 | 0.71 | 0.28 | 0.04 | 0.04 | 0.21 | 0.17 | 0.32 |
| **KVComm (0.7)** | **0.56** | **0.98** | **0.72** | **0.29** | **0.48** | **0.45** | **0.35** | **0.33** |
| $\mathcal{M}_s$: tiiuae/Falcon3-7B-Instruct; $\mathcal{M}_r$: tiiuae/Falcon3-7B-Instruct | | | | | | | | |
| **Random (0.3)** | 0.01 | 0.35 | 0.18 | 0.04 | 0.03 | 0.12 | 0.21 | 0.30 |
| **KVComm (0.3)** | **0.06** | **0.67** | **0.41** | **0.12** | **0.22** | **0.41** | **0.23** | **0.32** |
| **Random (0.5)** | 0.04 | 0.41 | 0.24 | 0.03 | 0.05 | 0.16 | 0.24 | 0.31 |
| **KVComm (0.5)** | **0.16** | **0.94** | **0.52** | **0.22** | **0.33** | **0.47** | **0.33** | **0.32** |

Table 9 – continued from previous page

| Method | Countries | Tipsheets | HotpotQA | QASPER | MuSiQuest | MultiField-QA-en | 2WikiM-QA | TMATH |
|---|---|---|---|---|---|---|---|---|
| **Random (0.7)** | 0.19 | 0.95 | 0.51 | 0.20 | 0.29 | 0.42 | 0.26 | **0.32** |
| **KVComm (0.7)** | **0.23** | **0.96** | **0.54** | **0.22** | **0.32** | **0.47** | **0.29** | **0.32** |
| $\mathcal{M}_s$: yuvraj17/EvolCodeLlama-3.1-8B-Instruct; $\mathcal{M}_r$: Team-ACE/ToolACE-2-Llama-3.1-8B | | | | | | | | |
| **Random (0.3)** | 0.00 | 0.34 | 0.06 | 0.00 | 0.01 | 0.13 | 0.03 | 0.34 |
| **KVComm (0.3)** | **0.12** | **0.95** | **0.12** | **0.05** | **0.04** | **0.26** | **0.19** | **0.36** |
| **Random (0.5)** | 0.03 | 0.79 | 0.29 | 0.06 | 0.09 | 0.32 | 0.16 | 0.35 |
| **KVComm (0.5)** | **0.55** | **0.98** | **0.38** | **0.15** | **0.14** | **0.43** | **0.28** | **0.38** |
| **Random (0.7)** | 0.37 | 0.85 | **0.59** | 0.21 | **0.27** | 0.47 | **0.33** | 0.36 |
| **KVComm (0.7)** | **0.53** | **0.97** | 0.51 | **0.22** | 0.25 | **0.49** | **0.33** | **0.38** |

# H    CALIBRATION SET SIZE

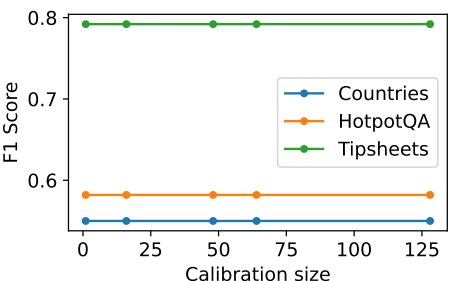

Figure 11: Effect of calibration set size. Calibration set size does not significantly affect the test performance.

We investigate how many samples are needed in the calibration set so that the selection strategy can generalize well to the test set. If a smaller calibration set can achieve good performance on the test set, it would be more practical since it would require less cost to obtain the selected layers. We conduct the experiment on Countries, Tipsheets, and HotpotQA datasets using the Llama-3.2-3B model. As the results in Figure 11 show, we can see that using only one sample in the calibration set can already achieve the same performance as using more samples (up to 128 samples). This suggests that our selection strategy can generalize well to the test set even with a very small calibration set. In all other experiments in the paper, we use one sample in the calibration set.

# I    IMPACT OF TRANSMITTED TOKEN LENGTH ON NLD

Transmitted token length is an important factor affecting the performance of natural language-based communication methods like NLD, which refers to the maximum number of tokens generated by the sender model to communicate with the receiver model. To investigate the impact of transmitted token length on NLD, we conduct experiments on HotpotQA, MultiFieldQA-en, and 2WikiMQA datasets with different transmitted token lengths ranging from 64 to 1024 tokens. The results are shown in Figure 12. We can see that as the transmitted token length increases from 64 to 128, the performance of NLD improves. However, as the transmitted token length continues to increase beyond 128 tokens, the performance gains become marginal. This suggests that there is a moderate transmitted token length (e.g., 128 tokens) is sufficient without incurring excessive communication overhead. In our main experiments, we set the transmitted token length to 256 tokens for NLD to ensure a fair comparison with other methods.

# J    MULTI-SOURCE KVCOMM

## J.1    EXTENDING KVCOMM TO MULTIPLE SOURCES

KVComm can be naturally extended to multiple sources by integrating the KV pairs from different sender models. Mathematically, if we have $N_s$ sender models $\mathcal{M}_{s_1}, \mathcal{M}_{s_2}, \ldots, \mathcal{M}_{s_{N_s}}$ and one receiver model $\mathcal{M}_r$, each sender $\mathcal{M}_{s_i}$ processes the context $C_i$ and generates its own KV pairs $\{(\mathbf{k}_{s_i}^l, \mathbf{v}_{s_i}^l)\}$ at each layer $l$. The receiver $\mathcal{M}_r$ can then receive the KV pairs from all senders and use them to compute the attention scores. The

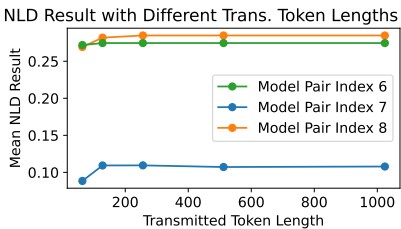

Figure 12: Effect of transmitted token length on NLD. A moderate length is sufficient for NLD.

attention scores can be computed as follows:

$$\hat{S}_a^l = \frac{1}{HT} \sum_{h=1}^{H} \sum_{q=1}^{|Q|} \sum_{i=1}^{N_s} \sum_{c=1}^{|C_i|} a_{h,q,i,c}^l,$$

where $|C_i|$ is the number of tokens in the context $C_i$, and $a_{h,q,i,c}^l$ is the attention weight assigned by head $h$ at layer $l$ from token $q$ to the context token $c$ of sender $\mathcal{M}_{s_i}$. The attention scores $\hat{S}_a^l$ are then integrated with the Gaussian prior to compute the selection scores.

Given the selection scores, a subset of KV pairs $\{(\mathbf{k}_{s_i}^{l_j}, \mathbf{v}_{s_i}^{l_j})\}$ can be selected from each sender model $\mathcal{M}_{s_i}$ at each layer $l_j$. The selected KV pairs are concatenated to form the final KV pairs for the receiver model $\mathcal{M}_r$:

$$\mathbf{k}_r^l \leftarrow [\mathbf{k}_{s_1}^{l_j}; \mathbf{k}_{s_2}^{l_j}; \ldots; \mathbf{k}_{s_{N_s}}^{l_j}; \mathbf{k}_r^l],$$
$$\mathbf{v}_r^l \leftarrow [\mathbf{v}_{s_1}^{l_j}; \mathbf{v}_{s_2}^{l_j}; \ldots; \mathbf{v}_{s_{N_s}}^{l_j}; \mathbf{v}_r^l].$$

where $l$ corresponds to a selected layer $l_j$.

### J.2 EXPERIMENT WITH TWO SENDERS AND ONE RECEIVER

We experiment with the scenario of two senders and one receiver to demonstrate the feasibility of extending KVComm to multiple sources. As shown in Table 10, we find that two senders can outperform one sender, for 17 out of 27 cases. We argue this is because of the diversification of information sources and agent thought. Owing to the usage of KV pairs, we can naturally integrate multiple sources, while NLD and CIPHER cannot, suffering performance degradation.

## K ADDITIONAL DISCUSSION

We have additional discussions on the details and choices of our method.

**Positional Embedding Coherence**  KVComm is designed to preserve positional coherence across all layers. For the receiver model, in each layer, we shift all its positions by $|C|$, where $|C|$ is the length of the context. For selected layers, we concatenate the KV pairs of the sender at positions $[0, |C|)$, and the KV pairs of the receiver follow at positions $[|C|, |C| + |Q|)$. For non-selected layers, positions $[0, |C|)$ are left empty (unattended), but the KV of the receiver still starts at position $|C|$. This approach ensures that all layers share a consistent positional frame, so the attention mechanism sees the same offsets at every depth, avoiding positional drift across layers. We perform an ablation study to validate this design in Appendix M.

**Communication Cost**  Under the scenario where agents are connected with high-bandwidth links, the communication cost is relatively low compared to recomputation cost (Jin et al., 2024; Liu et al., 2024c). KVComm is more preferred when the information exchange volume is large (e.g., long contexts) and the communication bandwidth is sufficient. In scenarios with limited bandwidth, further compression of KV pairs or more aggressive layer selection may be necessary, which we leave for future work.

## L CONTEXT-ADAPTIVE ONLINE CALIBRATION

A simple yet effective dynamic selection mechanism is to recompute the selected layers every $T$ queries, where $T$ is a hyperparameter that can be dynamically determined by server workload. We make two fully mixed

Table 10: Communication results for one sender and two senders scenarios. We **bold** the better result comparing one sender and two senders for each method. **KVComm (0.3/0.5/0.7)** denotes selecting 30%/50%/70% of layers' KV pairs for communication, i.e., $M = \lceil 0.3L \rceil$, $M = \lceil 0.5L \rceil$, $M = \lceil 0.7L \rceil$.

| Method | Sender | HotpotQA | MuSiQuest | 2WikiMQA |
|---|---|---|---|---|
| $\mathcal{M}_{s_1}$: **arcee-ai/Llama-3.1-SuperNova-Lite;** $\mathcal{M}_{s_2}$: **yuvraj17/EvolCodeLlama-3.1-8B-Instruct;** $\mathcal{M}_r$: **Team-ACE/ToolACE-2-Llama-3.1-8B** | | | | |
| Baseline | NA | 0.04 | 0.01 | 0.01 |
| Skyline | | 0.37 | 0.15 | 0.25 |
| NLD | $\mathcal{M}_{s_2}$ | **0.17** | **0.05** | 0.02 |
| | $\mathcal{M}_{s_1}$ and $\mathcal{M}_{s_2}$ | 0.14 | 0.04 | **0.03** |
| CIPHER | $\mathcal{M}_{s_2}$ | **0.19** | **0.06** | **0.03** |
| | $\mathcal{M}_{s_1}$ and $\mathcal{M}_{s_2}$ | 0.16 | 0.05 | **0.03** |
| KVComm (0.3) | $\mathcal{M}_{s_2}$ | 0.12 | 0.04 | 0.19 |
| | $\mathcal{M}_{s_1}$ and $\mathcal{M}_{s_2}$ | **0.16** | **0.06** | **0.23** |
| KVComm (0.5) | $\mathcal{M}_{s_2}$ | 0.38 | 0.14 | **0.28** |
| | $\mathcal{M}_{s_1}$ and $\mathcal{M}_{s_2}$ | **0.39** | **0.20** | 0.21 |
| KVComm (0.7) | $\mathcal{M}_{s_2}$ | 0.51 | 0.25 | 0.33 |
| | $\mathcal{M}_{s_1}$ and $\mathcal{M}_{s_2}$ | **0.53** | **0.29** | **0.34** |
| $\mathcal{M}_{s_1}$: **cooperleong00/Qwen2.5-7B-Instruct-Jailbroken;** $\mathcal{M}_{s_2}$: **Orion-zhen/Qwen2.5-7B-Instruct-Uncensored;** $\mathcal{M}_r$: **bespokelabs/Bespoke-Stratos-7B** | | | | |
| Baseline | NA | 0.13 | 0.03 | 0.09 |
| Skyline | | 0.53 | 0.25 | 0.09 |
| NLD | $\mathcal{M}_{s_2}$ | 0.16 | 0.04 | **0.02** |
| | $\mathcal{M}_{s_1}$ and $\mathcal{M}_{s_2}$ | **0.18** | **0.06** | **0.02** |
| CIPHER | $\mathcal{M}_{s_2}$ | **0.03** | **0.03** | **0.03** |
| | $\mathcal{M}_{s_1}$ and $\mathcal{M}_{s_2}$ | 0.02 | 0.01 | 0.01 |
| KVComm (0.3) | $\mathcal{M}_{s_2}$ | **0.02** | **0.01** | **0.08** |
| | $\mathcal{M}_{s_1}$ and $\mathcal{M}_{s_2}$ | **0.02** | 0.00 | 0.05 |
| KVComm (0.5) | $\mathcal{M}_{s_2}$ | 0.28 | 0.12 | **0.10** |
| | $\mathcal{M}_{s_1}$ and $\mathcal{M}_{s_2}$ | **0.32** | **0.18** | 0.08 |
| KVComm (0.7) | $\mathcal{M}_{s_2}$ | 0.41 | **0.25** | 0.15 |
| | $\mathcal{M}_{s_1}$ and $\mathcal{M}_{s_2}$ | **0.50** | 0.24 | **0.24** |
| $\mathcal{M}_{s_1}$: **RedaAlami/Falcon3-7B-Instruct-Distill-DS-v1;** $\mathcal{M}_{s_2}$: **ehristoforu/falcon3-ultraset;** $\mathcal{M}_r$: **huihui-ai/Falcon3-7B-Instruct-abliterated** | | | | |
| Baseline | NA | 0.21 | 0.04 | 0.23 |
| Skyline | | 0.76 | 0.56 | 0.45 |
| NLD | $\mathcal{M}_{s_2}$ | **0.52** | **0.25** | **0.24** |
| | $\mathcal{M}_{s_1}$ and $\mathcal{M}_{s_2}$ | 0.22 | 0.13 | 0.20 |
| CIPHER | $\mathcal{M}_{s_2}$ | **0.27** | **0.07** | **0.25** |
| | $\mathcal{M}_{s_1}$ and $\mathcal{M}_{s_2}$ | 0.15 | 0.04 | 0.14 |
| KVComm (0.3) | $\mathcal{M}_{s_2}$ | **0.59** | **0.40** | **0.29** |
| | $\mathcal{M}_{s_1}$ and $\mathcal{M}_{s_2}$ | 0.51 | 0.30 | 0.27 |
| KVComm (0.5) | $\mathcal{M}_{s_2}$ | **0.63** | **0.44** | 0.34 |
| | $\mathcal{M}_{s_1}$ and $\mathcal{M}_{s_2}$ | 0.49 | 0.43 | **0.35** |
| KVComm (0.7) | $\mathcal{M}_{s_2}$ | 0.55 | 0.42 | **0.31** |
| | $\mathcal{M}_{s_1}$ and $\mathcal{M}_{s_2}$ | **0.60** | **0.46** | **0.31** |

datasets, i.e., mixing all the samples from two datasets: Countries and Tipsheets; Countries and MultiFieldQA-en. We then perform online calibration and evaluate with different calibration intervals $T$. As shown in Figure 13, we find the performance drops when $T$ increases, which is consistent with intuition.

Beyond periodic recomputation, more sophisticated adaptive mechanisms are also feasible. For example, the receiver model could leverage lightweight signals, such as token-level entropy and attention sparsity patterns, to trigger on-demand re-selection of informative layers. This is an exciting direction for future work, and KVComm provides a clean foundation for such extensions.

Additionally, to illustrate how different the selected layers are for different datasets, we calculate the Kendall's Tau similarity of layer rankings for each pair of datasets across all models. As shown in Figure 14, some tasks share quite a similar layer ranking for a given model pair, e.g., model pair index 6 shares a similar layer ranking for HotpotQA and MuSiQuest datasets. This phenomenon could guide the design of dynamic selection mechanisms in future work.

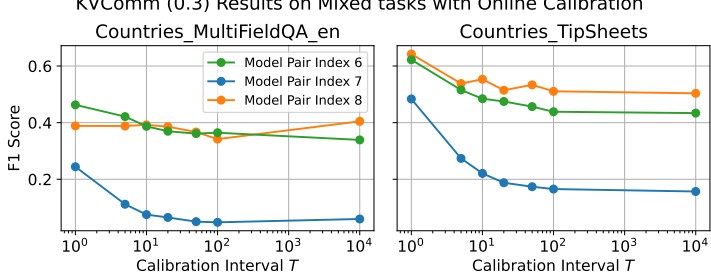

Figure 13: Online calibration performance drops when the calibration interval increases.

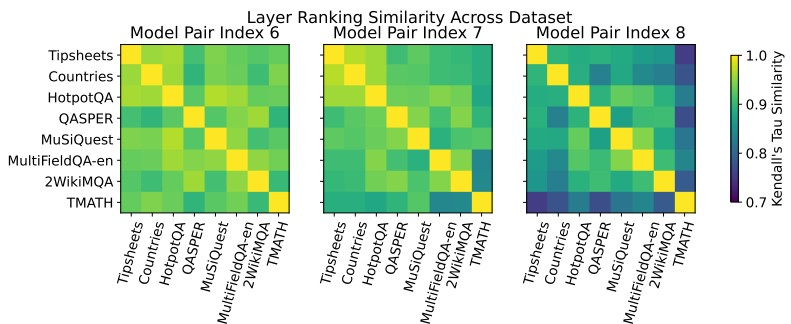

Figure 14: Kendall's Tau similarity of layer rankings between different datasets.

## M  POSITIONAL EMBEDDING COHERENCE

Table 11: Comparison of KVComm and KVComm-S. KVComm-S denotes shifting back the token positions of non-selected layers to 0. We bold the best results between KVComm and KVComm-S under the same settings.

| Method | Countries | Tipsheets | HotpotQA | QASPER | MuSiQuest | MultiField-QA-en | 2WikiM-QA | TMATH |
|---|---|---|---|---|---|---|---|---|
| $\mathcal{M}_s$: **huihui-ai/Llama-3.2-3B-Instruct-abliterated;** $\mathcal{M}_r$: **suayptalha/DeepSeek-R1-Distill-Llama-3B** | | | | | | | | |
| **KVComm-S (0.3)** | 0.26 | **0.65** | 0.40 | **0.10** | 0.14 | **0.19** | 0.21 | **0.36** |
| **KVComm (0.3)** | **0.46** | 0.45 | **0.46** | 0.09 | **0.28** | 0.15 | **0.28** | 0.35 |
| **KVComm-S (0.5)** | 0.49 | 0.74 | **0.57** | **0.28** | **0.32** | 0.45 | 0.30 | **0.35** |
| **KVComm (0.5)** | **0.57** | **0.81** | **0.57** | 0.27 | **0.32** | **0.51** | **0.36** | **0.35** |
| **KVComm-S (0.7)** | 0.52 | 0.76 | **0.65** | **0.30** | **0.39** | 0.46 | 0.32 | **0.35** |
| **KVComm (0.7)** | **0.57** | **0.81** | **0.65** | 0.29 | 0.36 | **0.47** | **0.37** | **0.35** |
| $\mathcal{M}_s$: **Orion-zhen/Qwen2.5-7B-Instruct-Uncensored;** $\mathcal{M}_r$: **bespokelabs/Bespoke-Stratos-7B** | | | | | | | | |
| **KVComm-S (0.3)** | 0.00 | 0.20 | **0.02** | **0.02** | **0.01** | **0.09** | 0.05 | **0.34** |
| **KVComm (0.3)** | **0.04** | **0.26** | **0.02** | 0.01 | 0.01 | **0.09** | **0.08** | 0.31 |
| **KVComm-S (0.5)** | 0.04 | **0.90** | **0.33** | **0.13** | **0.18** | **0.35** | **0.16** | **0.35** |
| **KVComm (0.5)** | **0.19** | 0.88 | 0.28 | 0.07 | 0.12 | 0.26 | 0.10 | 0.33 |
| **KVComm-S (0.7)** | 0.36 | **0.94** | **0.42** | 0.19 | 0.24 | **0.35** | **0.16** | **0.34** |
| **KVComm (0.7)** | **0.41** | 0.89 | 0.41 | **0.21** | **0.25** | 0.29 | 0.15 | **0.34** |
| $\mathcal{M}_s$: **ehristoforu/falcon3-ultraset;** $\mathcal{M}_r$: **huihui-ai/Falcon3-7B-Instruct-abliterated** | | | | | | | | |
| **KVComm-S (0.3)** | **0.47** | **0.71** | 0.54 | 0.10 | 0.36 | 0.19 | 0.26 | **0.32** |
| **KVComm (0.3)** | 0.46 | 0.69 | **0.59** | **0.19** | **0.40** | **0.35** | **0.29** | **0.32** |
| **KVComm-S (0.5)** | 0.36 | **0.97** | **0.67** | **0.27** | **0.46** | 0.34 | **0.36** | 0.34 |
| **KVComm (0.5)** | **0.40** | 0.92 | 0.63 | 0.25 | 0.44 | **0.45** | 0.34 | **0.35** |
| **KVComm-S (0.7)** | **0.21** | 0.95 | **0.59** | 0.26 | **0.52** | 0.46 | **0.37** | **0.36** |
| **KVComm (0.7)** | 0.19 | **0.96** | 0.55 | 0.26 | 0.42 | **0.51** | 0.31 | **0.36** |

We quantify the importance of positional embedding coherence between the sender and receiver models. We perform an ablation experiment where, for non-selected layers, instead of shifting the receiver's tokens to

position $|C|$, we place them back to position 0, creating a positional inconsistency with selected layers. As shown in Table 11, positional inconsistency does not have a detrimental effect on performance, but overall, our approach has merit.

# N  COMPLEXITY ANALYSIS DETAILS

We compare the computational complexity of our KVComm framework with the Skyline method and the NLD method. Recall that $L$ is the total number of layers in the model, $M$ is the number of selected layers for communication. We use $d$ to denote the hidden dimension of the model, and $|Q|$ and $|C|$ to denote the number of tokens in the query and context, respectively. Suppose $\mathcal{M}_r$ would generate $T_r$ tokens in total, and the number of generated tokens is the same across different methods. For NLD, $\mathcal{M}_s$ and $\mathcal{M}_r$ would each generate $T_s$ and $T_r$ tokens for the initial answer, respectively.

Ignoring the embedding, output layers, and other minor components, the computational complexity of prefilling a sequence of length $N$ with a single decoder layer is $O(Nd^2 + N^2d)$, while the complexity of decoding a single token is $O(d^2 + (N+i)d)$, where $i$ is the index of the generated token. Therefore, the total computational complexity of $\mathcal{M}_s$ to process the context $C$ is $O(L(|C|d^2 + |C|^2d))$.

The total computational complexity of KVComm consists of three parts: (1) the complexity of $\mathcal{M}_s$ to process the context $C$, which is $O(L(|C|d^2 + |C|^2d))$, (2) the complexity of $\mathcal{M}_r$ to process the query $Q$ with the selected $M$ KV pairs from $\mathcal{M}_s$, which is $O(L|Q|d^2 + M(|C| + |Q|)|Q|d + (L - M)|Q|^2d)$, and (3) the complexity of $\mathcal{M}_r$ to generate $T_r$ tokens with the selected $M$ KV pairs from $\mathcal{M}_s$, which is $O(T_r(Ld^2 + M(|C| + |Q| + T_r)d + (L - M)(|Q| + T_r)d))$. Therefore, the total computational complexity of KVComm is:

$$\mathcal{T}(\text{KVComm}) = O\Big(L\left(|C| + |Q| + T_r\right)d^2\Big)$$
$$+ O\Big(\big(L\left(|C|^2 + |Q|^2 + T_r^2 + T_r|Q|\right) + CM\left(|Q| + T_r\right)\big)d\Big)$$

The computational complexity of Skyline method consists of two parts: (1) the complexity of prefilling the concatenation of the context $C$ and query $Q$, which is $O(L(|C| + |Q|)d^2 + L(|C| + |Q|)^2d)$, and (2) the complexity of decoding $T_r$ tokens, which is $O(TL(d^2 + (|C| + |Q| + T_r)d))$. Therefore, the total computational complexity of the Skyline method is:

$$\mathcal{T}(\text{Skyline}) = O\Big(L(|C| + |Q| + T_r)d^2\Big)$$
$$+ O\Big(L\big((|C| + |Q|)^2 + T_r(|C| + |Q| + T_r)\big)d\Big)$$

The margin of KVComm over Skyline is:

$$\mathcal{T}(\text{Skyline}) - \mathcal{T}(\text{KVComm}) = O\Big(|C|d\big(L(2|Q| + T_r) - M(|Q| + T_r)\big)\Big)$$

For NLD, the total computational complexity consists of three parts: (1) the complexity of $\mathcal{M}_s$ to process the context $C$ and generate $T_s$ tokens, which is $O(L(|C|d^2 + |C|^2d) + T_sL(d^2 + (|C| + T_s)d))$, (2) the complexity of $\mathcal{M}_r$ to process the query $Q$ and generate $T_r$ tokens, which is $O(L(|Q|d^2 + |Q|^2d) + T_rL(d^2 + (|Q| + T_r)d))$, and (3) the complexity of $\mathcal{M}_r$ to process the entire debate history and generate $T_r$ tokens, which is $O(L((T_s + T_r + |Q|)d^2 + (T_s + T_r + |Q|)^2d) + TL(d^2 + (T_s + T_r + |Q| + T_r)d))$. Therefore, the total computational complexity of NLD is:

$$\mathcal{T}(\text{NLD}) = O\bigg(L\Big(|C| + 2|Q| + 2T_s + 2T_r + T_r\Big)d^2\bigg)$$
$$+ O\bigg(L\Big(|C|^2 + T_s^2 + |Q|^2 + T_r^2 + (T_s + T_r + |Q|)^2$$
$$+ T_r\big(T_s + T_r + T_r + |Q|\big) + T_s|C| + T_r|Q|\Big)d\bigg)$$

The margin of KVComm over NLD is:

$$
\begin{aligned}
\mathcal{T}(\text{NLD}) - \mathcal{T}(\text{KVComm}) = {} & O\Big(L\big(2T_s + 2T_r + |Q|\big)d^2\Big) \\
& + O\bigg(\bigg(L\Big(T_s^2 + T_r^2 + (T_s + T_r + |Q|)^2 \\
& + T_s|C| + T_r|Q| + T_r(T_s + T_r)\Big) - CM\big(|Q| + T_r\big)\bigg)d\bigg)
\end{aligned}
$$

## O  USING ONE CHUNK OF LAYERS

We conduct the same experiment as in Section 4.3 on the HotpotQA dataset using other model pairs in Table 5. The results are shown in Figure 15. We can see that the results are consistent with the observation in Section 4.3.

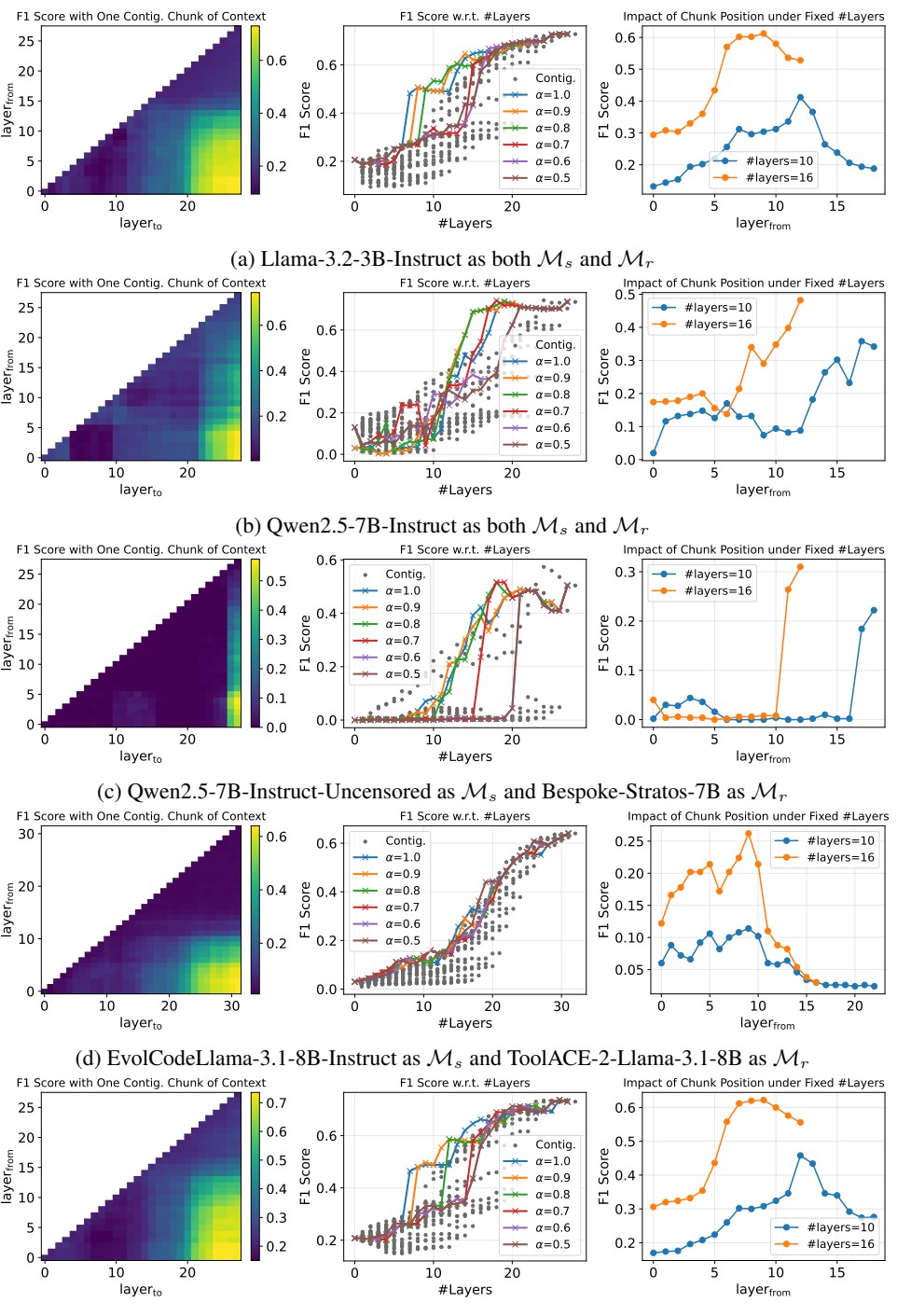

(a) Llama-3.2-3B-Instruct as both $\mathcal{M}_s$ and $\mathcal{M}_r$

(b) Qwen2.5-7B-Instruct as both $\mathcal{M}_s$ and $\mathcal{M}_r$

(c) Qwen2.5-7B-Instruct-Uncensored as $\mathcal{M}_s$ and Bespoke-Stratos-7B as $\mathcal{M}_r$

(d) EvolCodeLlama-3.1-8B-Instruct as $\mathcal{M}_s$ and ToolACE-2-Llama-3.1-8B as $\mathcal{M}_r$

(e) Llama-3.2-3B-Instruct-abliterated as $\mathcal{M}_s$ and DeepSeek-R1-Distill-Llama-3B as $\mathcal{M}_r$

Figure 15: Experiment results of using one chunk of layers for communication on HotpotQA dataset using different model pairs.

