# OpenReview forum: "KVComm: Enabling Efficient LLM Communication through Selective KV Sharing"
_ICLR.cc/2026/Conference — ICLR 2026 Poster_

### Official Review · Reviewer_XrXt · 2025-10-22

**Soundness:** 2
**Presentation:** 3
**Contribution:** 2
**Rating:** 4
**Confidence:** 3

**Summary:**

The paper proposes KVComm for inter-LLM communication via selective KV-cache sharing. The specific techinique is guided by attention-based importance and Gaussian layer prior. Experiments show moderate gains over prior activation- or embedding-based protocols, yet KVComm still sometimes fall short of the full transmission baselines.

**Strengths:**

- Selective KV-cache sharing with attention-based layer choice yields measurable gains over prior hidden-state or embedding protocols.
- Single-sample calibration generalizes across datasets, offering a lightweight deployment recipe.

**Weaknesses:**

1. I may misunderstand: the method appears restricted to two-agent pairs that share a parameter space (Table 5: same model or parent-offspring). If KVComm neither extends to >2 agents nor to heterogeneous backbones, its scope seems a bit narrow.

2. The authors should clarify whether KVComm generalizes to fully-heterogeneous, multi-LLM scenarios (differing hidden dimensions & vocabularies); otherwise the contribution risks being incremental and may overlap prior work [1].

3. Line 78 claims “using all tokens’ hidden states does not guarantee effective communication,” yet the same input-concatenation is later labeled “Skyline” and outperforms KVComm on Tipsheets/MuSiQuest (Table 1); this reads somehow contradictory.

4. Excluding Skyline from “best result” bolding in Table 1 warrants justification; its inclusion would contextualize absolute gains.

5. Fixing k % of layers manually is not adaptive; an automatic, data-driven or model-driven criterion to decide the subset size would strengthen practicality.

6. suayptalha/DeepSeek-R1-Distill-Llama-3B seems to be an unofficial version; the official releases are DeepSeek-R1-Distill-Llama-8B and DeepSeek-R1-Distill-Llama-70B. Using an unofficial checkpoint complicates reproducibility a bit; why not adopt the official distillations?

7. The authors may want to discuss the discrepancy and their unique contribution compared with [2], which seems to share similar motivation yet stronger adaptivity compared to this work.

---

[1] Dense Communication between Language Models

[2] KVCOMM: Online Cross-context KV-cache Communication for Efficient LLM-based Multi-agent Systems

**Questions:**

Please refer to Weakness

---

> ### Author Response · Authors · 2025-11-20
> **Response to Reviewer XrXt (Part 1)**
>
> We thank you for the constructive comments and the thoughtful reading of our work. Below, we respond to each concern in detail.
>
> ***Q1. I may misunderstand: the method appears restricted to two-agent pairs that share a parameter space (Table 5: same model or parent-offspring). If KVComm neither extends to >2 agents nor to heterogeneous backbones, its scope seems a bit narrow.***
>
> **A1.1. heterogeneous backbones**
>
> Thank you for raising this important question. We confirm this as a limitation. As part of our preliminary work, we did consider the case of communication across LLMs with embedding spaces. Our preliminary results were positive, but we decided to break down the problem in such a way that we first demonstrate the possibility of efficiently and effectively communicating across models. Having evaluated the effectiveness of transmitting hidden states, we focused on KV as the key source of information. Thus, we initially chose models that belong to the same architecture, because KV pair structures differ significantly across heterogeneous architectures. We leave the problem of KV alignment across heterogeneous architectures to future work, which may leverage latent space projection or learnt adapters. To address this comment, we explicitly added a paragraph in Section 6 of the revised manuscript.
>
>
> **A1.2. Multiple agent setting**
>
> KVComm can easily extend to multiple senders and multiple receivers, rather than one sender and one receiver.
>
> **One-to-many**: For multiple receivers, each receiver can have its own layer selection, which easily makes it adaptive. Thus, KVComm natively supports the one-to-many communication model.
>
> **Many-to-one**: To handle multiple senders, an approach that we have implemented and evaluated is to concatenate KV pairs from multiple resources. This can be used in combination with orthogonal works, which selectively keep tokens [3].
>
> To demonstrate the feasibility of our approach, we experiment with the scenario of two senders and one receiver.  As shown in the table below, we find that two senders outperform one sender for KVComm. We argue this is because of the diversification of information sources and agent thought. Owing to the usage of KV pairs, we can naturally integrate multiple sources, while NLD and CIPHER cannot, suffering performance degradation.
>
> **To summarize**, we have demonstrated the feasibility of a two-to-one style of multi-agent communication and shown that it can outperform the one-to-one case. We incorporated these discussions in Section 6 and the results in Appendix J of the revised manuscript.
>
>
> (We **bold** the better result comparing one sender and two senders for each method. To save space, we only put the result of one model pair. Please refer to Table 10 in the revised manuscript uploaded on 20 Nov for full results.)
> | Sender                       | Method           | HotpotQA | MuSiQuest | 2WikiMQA |
> |-----------------------------|------------------|----------|-----------|----------|
> **$\mathcal{M}_{s_1}$: arcee-ai/Llama-3.1-SuperNova-Lite**
> **$\mathcal{M}_{s_2}$: yuvraj17/EvolCodeLlama-3.1-8B-Instruct**
> **$\mathcal{M}_r$: Team-ACE/ToolACE-2-Llama-3.1-8B**
> | NA                          | **Baseline**     | 0.04     | 0.01      | 0.01     |
> | NA                          | **Skyline**      | 0.37     | 0.15      | 0.25     |
> | $\mathcal{M}_{s_2}$       | **NLD**          | **0.17** | **0.05**  | 0.02     |
> | $\mathcal{M}\_{s_1}$ and $\mathcal{M}\_{s_2}$ | **NLD**          | 0.14     | 0.04      | **0.03** |
> | $\mathcal{M}_{s_2}$       | **CIPHER**       | **0.19** | **0.06**  | **0.03** |
> | $\mathcal{M}\_{s_1}$ and $\mathcal{M}\_{s_2}$ | **CIPHER**       | 0.16     | 0.05      | **0.03** |
> | $\mathcal{M}_{s_2}$       | **KVComm (0.3)** | 0.12     | 0.04      | 0.19     |
> | $\mathcal{M}\_{s_1}$ and $\mathcal{M}\_{s_2}$ | **KVComm (0.3)** | **0.16** | **0.06**  | **0.23** |
> | $\mathcal{M}_{s_2}$       | **KVComm (0.5)** | 0.38     | 0.14      | **0.28** |
> | $\mathcal{M}\_{s_1}$ and $\mathcal{M}\_{s_2}$ | **KVComm (0.5)** | **0.39** | **0.20**  | 0.21     |
> | $\mathcal{M}_{s_2}$       | **KVComm (0.7)** | 0.51     | 0.25      | 0.33     |
> | $\mathcal{M}\_{s_1}$ and $\mathcal{M}\_{s_2}$ | **KVComm (0.7)** | **0.53** | **0.29**  | **0.34** |
>
>
>
> [3] CacheBlend: Fast Large Language Model Serving for RAG with Cached Knowledge Fusion
>
> ---
>
> Please note this response continues in the next comment.

---

> ### Author Response · Authors · 2025-11-20
> **Response to Reviewer XrXt (Part 2)**
>
> ***Q2. The authors should clarify whether KVComm generalizes to fully-heterogeneous, multi-LLM scenarios (differing hidden dimensions & vocabularies); otherwise the contribution risks being incremental and may overlap prior work [1].***
>
> **A2.1. whether KVComm generalizes to fully-heterogeneous, multi-LLM scenarios**
>
> Please refer to the responses A1.1 and A1.2 to the previous question.
>
> **A2.2. prior work [1].**
>
> Thank you for pointing out this work. We provide the clarification to distinguish KVComm from Dense Communication below.
>
> KVComm targets the problem setting of effective and efficient communication between **independently** operating LLM agents during inference. In contrast, Dense Communication focuses on constructing a trainable modular model by stripping multiple LLMs and training learnable edges to transform dense hidden states between models, which shares more similarity to [4].
>
> KVComm directly exchanges layer-wise KV pairs and introduces a novel importance-based layer-selection strategy, reducing communication cost by up to 70% while preserving performance. Dense Communication does not address selective communication, KV transfer, or inference-time communication efficiency. Thus, although both works fall under the broad category of non-textual inter-LLM communication, the design goals, mechanisms, and empirical focus differ substantially.
>
> [4] StitchLLM: Serving LLMs, One Block at a Time (ACL 2025)
>
>
> ---
>
> ***Q3. Line 78 claims “using all tokens’ hidden states does not guarantee effective communication,” yet the same input-concatenation is later labeled “Skyline” and outperforms KVComm on Tipsheets/MuSiQuest (Table 1); this reads somehow contradictory.***
>
> Thank you for pointing this out. In line 78, “using all tokens” does not refer to the Skyline approach. Skyline refers to the receiver directly processing the concatenation of the context $C$ and query $Q$, while no communication is involved. It serves as a comparative method to measure the information loss of other methods when communicating.
>
> “using all tokens” in line 78 refers to the experiment in Section 2.2.2, in which we prepend all tokens' hidden states from $\mathcal{M}\_s$ to $\mathcal{M}\_r$. A detailed procedure description of this experiment is as follows:
>
> Considering two models $\mathcal{M}\_s$ and $\mathcal{M}\_r$, each with $L$ layers, given $C$ and $Q$ as input, we run a partial forward pass of $\mathcal{M}\_s$ until layer $k$ to obtain the hidden state $\mathbf{H}\_k^{s}\in \mathbb{R}^{|C| \times d}$ for all tokens in $C$, where $|C|$ is the number of tokens in $C$, and $d$ is the hidden dimension. Another partial forward pass of $\mathcal{M}\_r$ is run until layer $j$ to obtain the hidden state $\mathbf{H}\_j^{r}\in \mathbb{R}^{|Q| \times d}$ for all tokens in $Q$, where $|Q|$ is the number of tokens in $Q$. We then modify the hidden states of $\mathcal{M}\_r$ at layer $j$ by prepending the hidden states from $\mathcal{M}\_s$ at layer $k$ as follows:
>
> $$
> \tilde{\mathbf{H}}_j^{r} =
> \begin{bmatrix}
> \mathbf{H}_k^{s} \\\\
> \mathbf{H}_j^{r}
> \end{bmatrix}
> $$
>
> We continue the forward pass from layer $j+1$ to layer $L$ using the modified hidden states $\tilde{\mathbf{H}}_j^{r}$ as input, and obtain the final output of the model. We evaluate the model's performance on the task with different layers $k$ and $j$.
>
> We added this detailed description to Appendix D of the revised manuscript uploaded on 20 Nov.
>
> ---
>
> ***Q4. Excluding Skyline from “best result” bolding in Table 1 warrants justification; its inclusion would contextualize absolute gains.***
>
> We appreciate the reviewer’s suggestion. We chose not to include Skyline in the “best result” bolding because Skyline is not a feasible communication protocol. It serves purely as an upper-bound estimate by allowing the receiver to fully re-encode the entire context. Our goal is to compare practical communication methods that do not give the receiver direct access to the full context.
>
> ---
>
> ***Q5. Fixing k % of layers manually is not adaptive; an automatic, data-driven or model-driven criterion to decide the subset size would strengthen practicality.***
>
> We thank the reviewer for this excellent suggestion. In our current experiments, the proportion of selected layers is treated as a hyperparameter, primarily to demonstrate the effectiveness and stability of our layer-selection strategy under different budget regimes. Introducing a data-driven or model-driven mechanism to automatically determine the optimal subset size and composition is a promising extension and we plan to incorporate such adaptive selection as part of our future work to further enhance practicality and deployment robustness of our approach.
>
> ---
>
> Please note this response continues in the next comment.

---

> ### Author Response · Authors · 2025-11-20
> **Response to Reviewer XrXt (Part 3)**
>
> ***Q6. suayptalha/DeepSeek-R1-Distill-Llama-3B seems to be an unofficial version; the official releases are DeepSeek-R1-Distill-Llama-8B and DeepSeek-R1-Distill-Llama-70B. Using an unofficial checkpoint complicates reproducibility a bit; why not adopt the official distillations?***
>
> We appreciate this feedback. We add experiments with the official DeepSeek-R1-Distill-Llama-8B version. As shown in the results below, we can confirm that all trends reported in our main paper remain consistent. KVComm continues to outperform other baselines under comparable communication budgets.
>
>
>
> | **Method** | Countries | Tipsheets | HotpotQA | QASPER | MuSiQuest | MultiField-QA-en | 2WikiM-QA | TMATH |
> |-----------|-----------|-----------|----------|--------|-----------|-------------------|-----------|-------|
> **$\mathcal{M}_s$: arcee-ai/Llama-3.1-SuperNova-Lite;**
> **$\mathcal{M}_r$: deepseek-ai/DeepSeek-R1-Distill-Llama-8B**
> | **Baseline**      | 0.07 | 0.30 | 0.11 | 0.01 | 0.03 | 0.09 | 0.16 | 0.23 |
> | **Skyline**       | 0.55 | 0.80 | 0.52 | 0.17 | 0.40 | 0.41 | 0.16 | 0.29 |
> | **NLD**           | 0.30 | 0.39 | 0.20 | 0.02 | 0.06 | 0.08 | **0.19** | 0.22 |
> | **CIPHER**        | _0.47_ | _0.71_ | 0.27 | 0.03 | 0.11 | 0.14 | 0.14 | 0.18 |
> | **AC (mean)**     | 0.00 | 0.31 | 0.08 | 0.02 | 0.02 | 0.09 | 0.14 | 0.25 |
> | **AC (replace)**  | 0.00 | 0.39 | 0.04 | 0.01 | 0.00 | 0.16 | _0.16_ | _0.28_ |
> | **AC (sum)**      | 0.02 | 0.34 | 0.07 | 0.02 | 0.02 | 0.08 | _0.16_ | 0.24 |
> | **KVComm (0.3)**  | 0.09 | 0.52 | 0.10 | 0.01 | 0.03 | 0.09 | 0.08 | _0.28_ |
> | **KVComm (0.5)**  | 0.41 | **0.76** | _0.33_ | _0.05_ | _0.21_ | _0.23_ | 0.09 | **0.29** |
> | **KVComm (0.7)**  | **0.53** | **0.76** | **0.47** | **0.12** | **0.28** | **0.31** | 0.14 | **0.29** |
>
> We included the numerical results in Appendix F of the revised manuscript uploaded on 20 Nov.
>
> ---
>
> ***Q7. The authors may want to discuss the discrepancy and their unique contribution compared with [2], which seems to share similar motivation yet stronger adaptivity compared to this work.***
>
> Thank you for pointing out this paper. In a nutshell, our work goes beyond this related work by: 1) enabling a different type of communication (where the receiver does not have access to the context), 2) making it possible to efficiently and **selectively** choose layers of KV pairs that will be transmitted, and 3) being able to work effectively across different models that are fine-tuned from one model. Next we discuss the differences in more detail and offer additional insights.
>
> The work in [2] is orthogonal to our approach in that it aims to reduce prefill latency of identical models by studying KV cache reuse across agents that process overlapping textual histories. It adjusts KV cache for shared content by referencing a pool of cached examples-termed anchors that store observed cache deviations under varying prefixes.
>
> In contrast, our work focuses on **what** information should be communicated between different LLMs that may receive totally different inputs and play different roles. We compare hidden states and KV pairs as communication media, and identify the limitations of hidden-state-based communication. Our work selectively shares KV pairs of non-contiguous layers based on attention importance scores and a Gaussian prior. This yields strong task performance while transmitting as few as 30% of layers’ KV pairs.
>
> Importantly, our method does not require overlapping prefixes and thus applies to settings (e.g., contextual QA) where agents observe complementary, but different, instead of identical content.
>
> Finally, our approach works across models that have been fine-tuned from one model. This is also the case for our additional result with a two-to-one communication model.
>
> We added a discussion in Section 5 related work of the revised manuscript uploaded on 20 Nov, to clarify these differences.

---

> ### Author Response · Authors · 2025-11-27
> **Executive summary of our response and seeking additional feedback**
>
> Dear reviewer,
>
> We hope you are doing well. To very briefly summarize our response: we have demonstrated the ability to perform two-to-one style communication, which will hopefully alleviate your concern about the lack of multi-agent support. We have also evaluated using official DeepSeek releases and clarified that we go beyond the contemporaneous work named similarly to ours that appeared at the most recent NeurIPS. We sincerely hope that this will help you in deciding whether to revise your score and we are very happy to hear if you have any additional feedback.

---

> > ### Comment · Reviewer_XrXt · 2025-11-28
> >
> > Thank you very much for your detailed response and for conducting the additional experiments. Due to the current system settings, I am unfortunately unable to update my score. However, I will make sure to clearly convey my revised assessment of the paper to the Area Chair. Thank you again for your work.

---

### Official Review · Reviewer_i79Z · 2025-10-29

**Soundness:** 3
**Presentation:** 2
**Contribution:** 2
**Rating:** 4
**Confidence:** 4

**Summary:**

The paper proposes a framework for communication between LLMs in multi-agent systems. Instead of using natural language or hidden states to exchange information, it enables models to communicate efficiently by selectively sharing KV pairs from their attention layers. The authors introduce a layer-wise KV selection strategy, identifying the most informative layers for transmission. Experiments across eight model pairs and multiple reasoning datasets show that KVComm achieves performance comparable to an upper-bound “Skyline” setup while reducing computation and communication costs.

**Strengths:**

* The paper tackles an important but underexplored question: how to design effective and efficient communication protocols between LLMs in multi-agent systems. Both the *effectiveness* and *efficiency* of inter-agent communication are crucial for scaling such systems, making this direction timely and meaningful.
* The authors conduct several preliminary experiments to justify their design choices, such as analyzing token importance and the limitations of hidden-state communication. These analyses help motivate the proposed selective KV sharing mechanism.
* The paper compares against a wide range of baselines and evaluates on diverse tasks and model pairs.

**Weaknesses:**

* The proposed setup essentially reduces the “multi-agent” interaction to one model encoding the context and another generating the response. And since the paper uses KV-cache as medium, this feels closer to a *single-model sparse attention* or *cache reuse* scenario rather than true multi-agent communication. Moreover, if we really consider this problem under the "multi-agent" context, then transmitting activations or KV caches between agents incurs substantially higher communication cost (floating-point tensors) than natural language (token IDs). If minimizing "transmission cost" is a goal, as stated in Line 127, the paper needs stronger justification for why direct context forwarding or lightweight textual communication is not preferable.
* Some explanations are vague or underspecified. For example, the term "*strong* attention distribution" seems not formal; Section 2.2.1’s experimental procedure (“remove or retain hidden states of specific tokens”) is poorly explained; and Section 2.2.2’s description of “prepending hidden states” is ambiguous, does it concatenate layer j's output from speaker model to the first position of layer k of receiver, and go through the remaining forward pass? Figures 3 also do not clearly define axes or mapping between sender and receiver layers.
* The NLD baseline performs unexpectedly poorly across datasets. It is unclear whether this is due to an inappropriate setup or prompt mismatch. If the NLD agents still follow a debate-style dialogue rather than an information-transfer style, the comparison may not be meaningful, since the objective differs fundamentally.
* A key challenge of activation- or KV-based communication is its poor transferability across models with different architectures or base weights. This greatly limits the generality of the proposed method. The current experiments only involve models fine-tuned from the same base, so the claim of broad applicability is not yet substantiated.
* Several figures suffer from layout or readability problems, e.g., overlap in Figure 1 (“Extract attention”), overly tight spacing in Figures 2–3, and Figures 4–6.

**Questions:**

See above

---

> ### Author Response · Authors · 2025-11-20
> **Response to Reviewer i79Z (Part 1)**
>
> We sincerely thank you for the constructive and insightful feedback. We address each concern point-by-point below.
>
> ***Q1. difference from single-model sparse attention or cache reuse, communication cost compared to natural language communication, why not context forwarding or textual communication***
>
> We appreciate this valid concern and would like to succinctly point out that our approach enables an efficient and effective, qualitatively different, style of communication. Next we clarify these main points.
>
>
>
> **A1.1. The difference between KVComm and single-model sparse attention or cache reuse.**
> KVComm is fundamentally different from single-model sparse attention or cache reuse. The reasons are as follows:
> -   **Different computational settings:** Sparse attention or cache reuse works with only one model processing a single long input. But KVComm involves two separate, distinct models performing non-overlapping computation on different inputs (context vs. query). Aligned with multi-agent literature (e.g., CAMEL[1], AutoGen[2]), KVComm allows agents to process and exchange this asymmetric information to collaborate on a task, which does not appear in single model settings.
> -   **Different objectives:** The main aim of sparse attention/cache reuse is to minimize computation cost. However, the objective of KVComm is to achieve effective and efficient information exchange among agents. KVComm considers **what** information is most effective and how to minimize the communication cost.
> -   **Non-shared parameters:** Cache reuse assumes the same model with identical parameters and architecture. In contrast, the senders and receivers for KVComm may differ, with even heterogeneous model architectures. This flexibility is absent in single-model inference.
>
> We incorporated these discussions in Section 5 related work of the revised manuscript uploaded on 20 Nov.
>
> [1] CAMEL: Communicative Agents for "Mind" Exploration of Large Language Model Society
>
> [2] ​​AutoGen: Enabling Next-Gen LLM Applications via Multi-Agent Conversations
>
>
> **A1.2. Communication cost compared to natural language communication.**
> We agree that transmitting KV is more expensive than natural language. However, we clarify that the comparison strongly depends on the system setting and information transfer requirements.
>
> **High bandwidth deployment setting**. Our primary scenario uses fine-tuned open-source models hosted on GPU servers connected via high-speed links, e.g., 100 Gbps and beyond. Prior work [3,4] shows that in such a setting, KV cache reuse is more efficient than re-prefilling, which is required by natural language communication.
>
> **Decoding overhead by natural language communication**. Apart from the communication cost, natural language communication requires multiple decoding steps from the sender, which introduces substantial computational overhead and latency. KVComm avoids this overhead entirely by transmitting activations only.
>
> **Different suitability for different problem settings**. Natural language communication is more suitable when the amount of information to transmit is small or the sender already knows exactly what needs to be conveyed. In contrast, KVComm is more preferable when the information volume is large or the sender does not know about the query, and thus cannot efficiently summarize the context using natural language alone.
>
> **Communication cost of KVComm can be further reduced**. Given the flexibility of the KVComm framework, KV cache compression and quantization techniques can be integrated into it, which can be further investigated in future work.
>
> We added these discussions in Appendix K to the revised manuscript.
>
> [3] Compute Or Load KV Cache? Why not Both?
>
> [4] CacheGen: KV Cache Compression and Streaming for Fast Large Language Model Serving
>
>
> **A1.3. Why direct context forwarding or lightweight textual communication is not preferable.**
> We would like to clarify that the goal of KVComm is not to minimize communication cost alone, but to jointly optimize communication cost, computation cost, and information fidelity between agents.
>
> Although textual communication has low communication cost, it incurs **high computation cost** since senders must perform multiple rounds of computationally expensive decoding, and receivers have to re-prefill the information sent. For direct context forwarding, our experimental results in Section 2.2 demonstrate that it suffers from information loss, leading to **performance degradation**.
>
> Based on the above, we believe that direct context forwarding or lightweight textual communication is not preferable. We added these discussions to Section 2.1 of the revised manuscript.
>
> ---
>
> Please note this response continues in the next comment.

---

> ### Author Response · Authors · 2025-11-20
> **Response to Reviewer i79Z (Part 2)**
>
> ***Q2. Some explanations are vague or underspecified. For example, the term "strong attention distribution" seems not formal; Section 2.2.1’s experimental procedure (“remove or retain hidden states of specific tokens”) is poorly explained; and Section 2.2.2’s description of “prepending hidden states” is ambiguous, does it concatenate layer j's output from speaker model to the first position of layer k of receiver, and go through the remaining forward pass? Figures 3 also do not clearly define axes or mapping between sender and receiver layers.***
>
> **A2.1 Term "strong attention distribution"**
>
> We acknowledge that the phrasing was informal. We quantify this notion of *attention distribution* using the attention importance score $S_a^l$:
>
> $$\hat{S}\_a^l = \frac{1}{HT} \sum\_{h=1}^{H} \sum\_{t=1}^{T} \sum\_{c=1}^{C} a_{h,t,c}^l,$$
>
> where $H$ is the number of attention heads, $T$ is the number of tokens in the query, $C$ is the number of context tokens, and $a_{h,t,c}^l$ is the attention weight assigned by head $h$ at layer $l$ from token $t$ to context token $c$.
>
> We deem layer $l_i$ to exhibit stronger attention distribution than $l_j$, if $S_a^{l_i} > S_a^{l_j}$.
>
> We explicitly defined this term in Section 3.2 of the revised manuscript uploaded on 20 Nov.
>
> **A2.2 Section 2.2.1’s experimental procedure**
>
> Thank you for pointing out this issue. The following is a clearer procedure description of this experiment:
>
> Considering a model $\mathcal{M}$ with $L$ layers, given an input $X$ with $N$ tokens, we run a partial forward pass until layer $l$ to obtain the hidden states $\\{\mathbf{h}\_i^l\\}\_{i=1}^N$. Then, given a specific token position $k$, if we perform the **Retain** operation, we create a modified set of hidden states $\\{\tilde{\mathbf{h}}\_i^l\\}\_{i=1}^N$ as follows:
>
> $$
> \tilde{\mathbf{h}}\_i^l =
> \begin{cases}
> \mathbf{h}\_i^l, & \text{if } i = k \\\\
> \mathbf{0}, & \text{otherwise}
> \end{cases}
> $$
>
> If we perform the **Remove** operation, we create the modified set of hidden states $\\{\tilde{\mathbf{h}}\_i^l\\}\_{i=1}^N$ as follows:
>
> $$
> \tilde{\mathbf{h}}\_i^l =
> \begin{cases}
> \mathbf{0}, & \text{if } i = k \\\\
> \mathbf{h}\_i^l, & \text{otherwise}
> \end{cases}
> $$
>
> We then continue the forward pass from layer $l+1$ to layer $L$ using the modified hidden states $\\{\tilde{\mathbf{h}}\_i^l\\}\_{i=1}^N$ as input, and obtain the final output of the model. We evaluate the model's performance on the task with different token positions $k$ and layer $l$.
>
> We added this detailed description to Appendix C of the revised manuscript uploaded on 20 Nov.
>
>
> **A2.3 Section 2.2.2’s description of “prepending hidden states”**
>
> We confirm that the reviewer’s interpretation is correct. Below is a clearer description of this experiment:
>
> Considering two models $\mathcal{M}_s$ and $\mathcal{M}_r$, each with $L$ layers, given $C$ and $Q$ as input, we run a partial forward pass of $\mathcal{M}_s$ until layer $k$ to obtain the hidden state $\mathbf{H}_k^{s}\in \mathbb{R}^{|C| \times d}$ for all tokens in $C$, where $|C|$ is the number of tokens in $C$, and $d$ is the hidden dimension. Another partial forward pass of $\mathcal{M}_r$ is run until layer $j$ to obtain the hidden state $\mathbf{H}_j^{r}\in \mathbb{R}^{|Q| \times d}$ for all tokens in $Q$, where $|Q|$ is the number of tokens in $Q$. We then modify the hidden states of $\mathcal{M}_r$ at layer $j$ by prepending the hidden states from $\mathcal{M}_s$ at layer $k$ as follows:
>
> $$
> \tilde{\mathbf{H}}_j^{r} =
> \begin{bmatrix}
> \mathbf{H}_k^{s} \\\\
> \mathbf{H}_j^{r}
> \end{bmatrix}
> $$
>
> We continue the forward pass from layer $j+1$ to layer $L$ using the modified hidden states $\tilde{\mathbf{H}}_j^{r}$ as input, and obtain the final output of the model. We evaluate the model's performance on the task with different layers $k$ and $j$.
>
> We added this detailed description to Appendix D of the revised manuscript uploaded on 20 Nov.
>
> **A2.4 Axes in Figure 3**
>
> The X-axis of Figure 3 is Receiver $\mathcal{M}_r$ layer index (j), while the Y-axis is Sender $\mathcal{M}_s$ layer index (k). We updated Figure 3 with axis labels and mapping notation in the revised manuscript uploaded on 20 Nov.
>
> ---
>
> Please note this response continues in the next comment.

---

> ### Author Response · Authors · 2025-11-20
> **Response to Reviewer i79Z (Part 3)**
>
> ***Q3. The NLD baseline performs unexpectedly poorly across datasets. It is unclear whether this is due to an inappropriate setup or prompt mismatch. If the NLD agents still follow a debate-style dialogue rather than an information-transfer style, the comparison may not be meaningful, since the objective differs fundamentally.***
>
> In summary, our code had a minor bug in the receiver-side prompt construction, but even after we fixed it, KVComm still outperforms NLD, which consistently uses the “information-transfer” style rather than the debate-style objective.
>
> **A3.1. The NLD baseline performs unexpectedly poorly across datasets.**
>
> We thank the reviewer for pointing out the unexpected performance patterns of the NLD baselines. After re-checking our implementation, we identified a minor bug in the receiver-side prompt construction, where the template was not correctly applied for the last round. We have fixed this issue and provide the revised table below:
>
>
> | **Method** | **Countries** | **Tipsheets** | **HotpotQA** | **QASPER** | **MuSiQuest** | **MultiField-QA-en** | **2WikiM-QA** | **TMATH** |
> |-----------|---------------|---------------|--------------|------------|----------------|-----------------------|----------------|-----------|
> **$\mathcal{M}_s$: huihui-ai/Llama-3.2-3B-Instruct-abliterated;**
> **$\mathcal{M}_r$: suayptalha/DeepSeek-R1-Distill-Llama-3B**
> | Baseline      | 0.05 | 0.32 | 0.23 | 0.05 | 0.02 | 0.11 | 0.27 | 0.34 |
> | Skyline       | 0.57 | 0.91 | 0.73 | 0.25 | 0.51 | 0.47 | 0.40 | 0.36 |
> | NLD           | 0.43 | _0.72_ | 0.43 | 0.10 | 0.18 | 0.09 | 0.30 | 0.33 |
> | CIPHER        | 0.42 | 0.69 | 0.50 | 0.10 | 0.18 | 0.13 | 0.32 | 0.32 |
> | KVComm (0.3)  | _0.46_ | 0.45 | 0.46 | 0.09 | 0.28 | 0.15 | 0.28 | **0.35** |
> | KVComm (0.5)  | **0.57** | **0.81** | _0.57_ | _0.27_ | _0.32_ | **0.51** | _0.36_ | **0.35** |
> | KVComm (0.7)  | **0.57** | **0.81** | **0.65** | **0.29** | **0.36** | _0.47_ | **0.37** | **0.35** |
> **$\mathcal{M}_s$: Orion-zhen/Qwen2.5-7B-Instruct-Uncensored;**
> **$\mathcal{M}_r$: bespokelabs/Bespoke-Stratos-7B**
> | Baseline      | 0.01 | 0.36 | 0.13 | 0.05 | 0.03 | 0.08 | 0.09 | 0.35 |
> | Skyline       | 0.51 | 0.97 | 0.53 | 0.10 | 0.25 | 0.40 | 0.09 | 0.35 |
> | NLD           | _0.21_ | 0.80 | 0.16 | 0.02 | 0.04 | 0.11 | 0.02 | **0.35** |
> | CIPHER        | 0.04 | 0.60 | 0.03 | 0.01 | 0.03 | 0.07 | 0.03 | _0.34_ |
> | KVComm (0.3)  | 0.04 | 0.26 | 0.02 | 0.01 | 0.01 | 0.09 | 0.08 | 0.31 |
> | KVComm (0.5)  | 0.19 | _0.88_ | _0.28_ | _0.07_ | _0.12_ | _0.26_ | _0.10_ | 0.33 |
> | KVComm (0.7)  | **0.41** | **0.89** | **0.41** | **0.21** | **0.25** | **0.29** | **0.15** | _0.34_ |
> **$\mathcal{M}_s$: ehristoforu/falcon3-ultraset;**
> **$\mathcal{M}_r$: huihui-ai/Falcon3-7B-Instruct-abliterated**
> | Baseline     | 0.08 | 0.36 | 0.21 | 0.06 | 0.04 | 0.09 | 0.23 | 0.31 |
> | Skyline      | 0.56 | 0.95 | 0.76 | 0.32 | 0.56 | 0.51 | 0.45 | 0.37 |
> | NLD          | **0.46** | 0.80 | 0.52 | 0.19 | 0.25 | 0.11 | 0.24 | 0.15 |
> | CIPHER       | 0.30 | 0.19 | 0.27 | 0.02 | 0.07 | 0.06 | 0.25 | 0.17 |
> | KVComm (0.3) | **0.46** | 0.69 | _0.59_ | 0.19 | 0.40 | 0.35 | 0.29 | 0.32 |
> | KVComm (0.5) | _0.40_ | _0.92_ | **0.63** | _0.25_ | **0.44** | _0.45_ | **0.34** | _0.35_ |
> | KVComm (0.7) | 0.19 | **0.96** | 0.55 | **0.26** | _0.42_ | **0.51** | _0.31_ | **0.36** |
>
> Please refer to **Table 1** and Appendix F for the full results of AC and other model pairs in the revised manuscript.
>
> Importantly, the overall trends remain unchanged. We do find that on Countries and Tipsheets datasets, NLD/CIPHER can achieve performance close to that of KVComm or Skyline. These datasets require only a very small and highly salient amount of information to be transferred. The sender’s responses in NLD/CIPHER happen to contain enough signal even without access to the question.
>
> However, for all other datasets, the sender has access to the entire context but not the question, and natural language communication cannot reliably extract and transmit the task-relevant subset of information. As a result, even after fixing the prompt template, NLD/CIPHER continues to perform substantially below KVComm on complex, long-context reasoning tasks.
>
> Thus, the corrected results continue to support our main conclusion.
>
> Moreover, we set a fixed communication length as the maximum number of tokens that can be generated for each round of “discussion”. We evaluate NLD on three datasets (HotpotQA, MultiFieldQA-en, and 2WikiMQA) with different lengths. As shown in **Figure 12** of the revised manuscript uploaded on 20 Nov, the transmitted token length does not have a huge impact on the results once it reaches a certain value, which is consistent with our explanation above.
>
> We updated the corresponding table and included discussions and results in Section 4.2 and Appendix I of the revised manuscript uploaded on 20 Nov.
>
>
> Please note that the responses to this question and the following questions continues in the next comment.

---

> ### Author Response · Authors · 2025-11-20
> **Response to Reviewer i79Z (Part 4)**
>
> **A3.2. Debate-style dialogue rather than an information-transfer style.**
>
> We agree with the reviewer that the original NLD is designed for debate and not for unidirectional information transfer. For this reason, in our implementation, we explicitly prompt the sender to summarize the context, ensuring that NLD is used in an “information-transfer” style rather than in a debate-style objective (details in Appendix B.4). Thus, the comparison is not between KVComm and standard debate, but between KVComm and the best possible natural language information-passing achievable within NLD.
>
> We added a more detailed discussion in Appendix B.4 of the revised manuscript uploaded on 20 Nov.
>
> ---
>
> ***Q4. A key challenge of activation- or KV-based communication is its poor transferability across models with different architectures or base weights. This greatly limits the generality of the proposed method. The current experiments only involve models fine-tuned from the same base, so the claim of broad applicability is not yet substantiated.***
>
> Thank you for raising this important question. We confirm this as a limitation. As part of our preliminary work, we did consider the case of communication across LLMs with embedding spaces. Our preliminary results were positive, but we decided to break down the problem in such a way that we first demonstrate the possibility of efficiently and effectively communicating across models. Having evaluated the effectiveness of transmitting hidden states, we focused on KV as the key source of information. Thus, we initially chose models that belong to the same architecture, because KV pair structures differ significantly across architectures. We leave the problem of cross-architecture KV alignment to future work, which may leverage latent space projection or learnt adapters. To address this comment, we explicitly added a paragraph in Section 6 of the revised manuscript uploaded on 20 Nov.
>
> ---
>
> ***Q5. Several figures suffer from layout or readability problems, e.g., overlap in Figure 1 (“Extract attention”), overly tight spacing in Figures 2–3, and Figures 4–6.***
>
> We appreciate this feedback and have corrected these issues in the revised manuscript uploaded on 20 Nov.

---

> ### Author Response · Authors · 2025-11-27
> **Executive summary of our response and seeking additional feedback**
>
> Dear reviewer,
>
> We hope you are doing well. To very briefly summarize our response: we have already demonstrated the qualitative benefit of communicating carefully selected KV pairs. Moreover, we have implemented a two-to-one communication scheme that shows an additional benefit from including more sending agents (Appendix J, Page 23, Table 10). Finally, we have found and fixed a bug in our NLD evaluation, which does not fundamentally change the experimental results. We sincerely hope that this will help you in deciding whether to revise your score and we are very happy to hear if you have any additional feedback.

---

### Official Review · Reviewer_1XmS · 2025-11-01

**Soundness:** 3
**Presentation:** 3
**Contribution:** 3
**Rating:** 8
**Confidence:** 4

**Summary:**

The paper introduces **KVComm**, a novel communication protocol for multi-agent systems based on Large Language Models (LLMs). Instead of relying on natural language or hidden states, KVComm selectively shares key-value (KV) pairs across models using a strategy guided by attention importance scores and a Gaussian prior. The method significantly reduces communication and computation overhead while maintaining or surpassing task performance on diverse benchmarks.

**Strengths:**

S1: Using KV pairs rather than natural language or hidden states is a compelling, less explored direction that avoids known bottlenecks like information loss and concentration bias.

S2: The use of attention importance with a Gaussian prior to guide layer selection is well-motivated and supported by empirical validation.

S3: The authors benchmark across multiple datasets and LLM pairs, including ablations and comparisons with strong baselines like Skyline and AC.

S4: The approach offers substantial computation and communication savings (2.5x–6x), making it relevant for scalable deployment scenarios.

S5: The paper provides detailed setup information, dataset samples, and makes its code and synthetic datasets available.

**Weaknesses:**

To be honest, I carefully read this paper but I have not found some serious weakness. Below are just some suggestions that may lead to a more solid work.



Suggest 1: The method is primarily validated on same or fine-tuned model pairs. Scalability to truly heterogeneous models remains unexplored.

Suggest 2: The selection strategy is static post-calibration. Dynamic or context-aware layer selection could further enhance flexibility.

Suggest 3: Several datasets are synthetically created or downsampled, which might not fully reflect real-world task complexity or communication needs.

**Questions:**

1. **How transferable is the KVComm strategy across model families?** The paper evaluates KVComm mostly on pairs of identical or fine-tuned models. How would this framework perform if the sender and receiver are *structurally different models*, e.g., GPT-style vs. Mistral-style? Would attention alignment or KV dimensions cause integration issues?

2. **Can the selection of layers be made context-adaptive instead of fixed?**
   The Gaussian prior is fixed during calibration. Could performance improve if KV layer selection is dynamically adjusted based on the current query or context (e.g., using entropy or other uncertainty measures)?

3. **Why use a Gaussian prior for layer selection?**
   While intuitive and empirically justified, the Gaussian prior assumes a unimodal distribution over informative layers. Have other priors or non-parametric approaches (e.g., entropy-weighted or data-driven learned selection) been considered?

4. **Does the integration of KV pairs affect positional encoding or cause mismatch issues?**
   Since KV pairs include positional biases (especially in rotary or absolute encodings), how does the concatenation affect the receiver’s decoding stability and accuracy?

5. **What is the impact on latency and memory footprint during runtime?**
   The paper discusses FLOPs and communication savings, but does the selective KV sharing induce memory overhead (e.g., extra caching or tensor reshaping), particularly for large batch sizes or streaming inference?

---

> ### Author Response · Authors · 2025-11-20
> **Response to Reviewer 1XmS (Part 1)**
>
> We sincerely thank you for the positive assessment and constructive suggestions. Below, we address each suggestion and question in detail.
>
>
> ***Suggest 1. The method is primarily validated on same or fine-tuned model pairs. Scalability to truly heterogeneous models remains unexplored. & Q1. How transferable is the KVComm strategy across model families?***
>
> Thank you for raising this important question. We confirm this as a limitation. As part of our preliminary work, we did consider the case of communication across LLMs with embedding spaces. Our preliminary results were positive, but we decided to break down the problem in such a way that we first demonstrate the possibility of efficiently and effectively communicating across models. Having evaluated the effectiveness of transmitting hidden states, we focused on KV as the key source of information. Thus, we initially chose models that belong to the same architecture, because KV pair structures differ significantly across heterogeneous architectures. We leave the problem of KV alignment across heterogeneous architectures to future work, which may leverage latent space projection or learnt adapters. To address this comment, we explicitly added a paragraph in Section 6 of the revised manuscript.
>
> ---
>
> ***Suggest 2: The selection strategy is static post-calibration. Dynamic or context-aware layer selection could further enhance flexibility. & Q2. Can the selection of layers be made context-adaptive instead of fixed?***
>
> We appreciate this suggestion. While KVComm currently adopts a fixed layer-selection strategy after calibration for simplicity and computational efficiency, we agree that context-adaptive selection is a promising extension. KVComm can naturally support online and dynamic selection. There is only a slight overload to perform the layer selection algorithm (only need prefill, no decoding), and one calibration example yields robust generalization.
>
>
> A simple yet effective dynamic selection mechanism is to recompute the selected layers every $T$ queries, where $T$ is a hyperparameter that can be dynamically determined by server workload. We make two fully mixed datasets and evaluate with different calibration intervals $T$. As shown in **Figure 13** of the revised manuscript uploaded on 20 Nov, we find the performance drops when $T$ increases, which is consistent with intuition.
>
> Beyond periodic recomputation, more sophisticated adaptive mechanisms are also feasible. For example, the receiver model could leverage lightweight signals, such as token-level entropy, attention sparsity patterns, to trigger on-demand re-selection of informative layers. This is an exciting direction for future work, and KVComm provides a clean foundation for such extensions.
>
> Additionally, to illustrate how different the selected layers are for different datasets, we calculate the Kendall's Tau similarity of layer rankings for each pair of datasets across all models. As shown in **Figure 14** of the revised manuscript uploaded on 20 Nov, some tasks share quite a similar layer ranking for a given model pair, e.g., model pair index 6 shares a similar layer ranking for HotpotQA and MuSiQuest datasets. This phenomenon could guide the design of dynamic selection mechanisms in future work.
>
> We incorporated these discussions in Section 6 and the results in Appendix L into the revised manuscript.
>
>
> Please note this response continues in the next comment.

---

> ### Author Response · Authors · 2025-11-20
> **Response to Reviewer 1XmS (Part 2)**
>
> ***Suggest 3: Several datasets are synthetically created or downsampled, which might not fully reflect real-world task complexity or communication needs.***
>
> Thank you for pointing this out. To address this concern, we expand three of our existing datasets (HotpotQA, QASPER, and MuSiQuest) to their full variants (HotpotQA-E, QASPER-E, and MuSiQuest-E), and we additionally include a new human-created summarization dataset, SAMSum, which represents a different task type and better reflects real-world multi-agent communication scenarios.
>
> As shown in the table below, the performance of KVComm on these larger and more realistic datasets is consistent with our previously reported results, demonstrating that our method generalizes beyond the synthetic or downsampled settings.
>
> (The table is formatted in a way that it first shows the source and receiver model pair, followed by results. There are multiple model pairs.)
> | **Method** | **HotpotQA-E** | **QASPER-E** | **MuSiQuest-E** | **SAMSum** |
> |-----------|----------------|--------------|------------------|------------|
> | **$\mathcal{M}_s$: huihui-ai/Llama-3.2-3B-Instruct-abliterated;**  |  |  |  |  |
> | **$\mathcal{M}_r$: suayptalha/DeepSeek-R1-Distill-Llama-3B** | | | | |
> | Baseline | 0.22 | 0.03 | 0.06 | 0.26 |
> | Skyline | 0.77 | 0.52 | 0.25 | 0.33 |
> | NLD | 0.45 | 0.10 | 0.18 | _0.28_ |
> | CIPHER | 0.51 | 0.10 | 0.20 | _0.28_ |
> | AC (mean) | 0.24 | 0.03 | 0.06 | 0.26 |
> | AC (replace) | 0.06 | 0.00 | 0.01 | 0.26 |
> | AC (sum) | 0.23 | 0.03 | 0.06 | 0.26 |
> | KVComm (0.3) | 0.44 | 0.25 | 0.11 | 0.25 |
> | KVComm (0.5) | _0.61_ | _0.36_ | _0.25_ | _0.28_ |
> | KVComm (0.7) | **0.71** | **0.38** | **0.30** | **0.29** |
> | **$\mathcal{M}_s$: Orion-zhen/Qwen2.5-7B-Instruct-Uncensored;** | | | | |
> | **$\mathcal{M}_r$: bespokelabs/Bespoke-Stratos-7B** | | | | |
> | Baseline | 0.15 | 0.04 | 0.06 | 0.25 |
> | Skyline | 0.58 | 0.27 | 0.10 | 0.35 |
> | NLD | _0.24_ | 0.02 | 0.07 | 0.28 |
> | CIPHER | 0.04 | 0.01 | 0.02 | **0.37** |
> | AC (mean) | 0.03 | 0.00 | 0.00 | 0.01 |
> | AC (replace) | 0.00 | 0.00 | 0.00 | 0.00 |
> | AC (sum) | 0.03 | 0.00 | 0.00 | 0.04 |
> | KVComm (0.3) | 0.02 | 0.00 | 0.01 | 0.18 |
> | KVComm (0.5) | 0.21 | _0.14_ | _0.08_ | 0.30 |
> | KVComm (0.7) | **0.40** | **0.34** | **0.21** | _0.35_ |
> | **$\mathcal{M}_s$: Orion-ehristoforu/falcon3-ultraset;** | | | | |
> | **$\mathcal{M}_r$: huihui-ai/Falcon3-7B-Instruct-abliterated** | | | | |
> | Baseline | 0.21 | 0.06 | 0.06 | 0.27 |
> | Skyline | 0.78 | 0.60 | 0.33 | 0.36 |
> | NLD | _0.52_ | 0.10 | 0.28 | 0.28 |
> | CIPHER | 0.28 | 0.03 | 0.09 | 0.17 |
> | AC (mean) | 0.24 | 0.06 | 0.07 | 0.26 |
> | AC (replace) | 0.12 | 0.02 | 0.01 | 0.26 |
> | AC (sum) | 0.23 | 0.05 | 0.06 | 0.26 |
> | KVComm (0.3) | **0.59** | 0.15 | 0.40 | 0.28 |
> | KVComm (0.5) | **0.59** | _0.22_ | **0.46** | _0.31_ |
> | KVComm (0.7) | **0.59** | **0.26** | _0.36_ | **0.32** |
>
> We also show the statistics of these datasets in the table below.
>
> | **Dataset** | **Size** |
> |-------------|-----------|
> | HotpotQA-E (Yang et al., 2018) | 7,405 |
> | QASPER-E (Dasigi et al., 2021) | 1,726 |
> | MuSiQuest-E (Trivedi et al., 2022) | 2,417 |
> | SAMSum (Gliwa et al., 2019) | 819 |
>
> We incorporated these expanded evaluations into Appendix E of the revised version to strengthen the empirical evidence.
>
> ---
>
> ***Q3. Why use a Gaussian prior for layer selection?***
>
> Thank you for raising this insightful point. We chose a Gaussian prior mainly for the following two reasons:
> * Prior work [1, 2] (Section 3.2) and our experiment results (Section 4.3) consistently show that layer informativeness exhibits a smooth, approximately unimodal trend, with a peak around the middle layers. This makes a Gaussian prior a simple yet effective inductive prior.
> * A parametric prior allows us to regularize the attention weight scores and avoid noisy selections. A Gaussian prior leads to a stable selection mechanism while adding two interpretable parameters $\mu$ and $\sigma$.
>
> Alternatives such as entropy-weighted or data-driven are promising but introduce significantly higher complexity, e.g., larger calibration sets, training a selector, or risking overfitting to a particular task distribution. Given our goal of keeping the method simple, efficient, and broadly reproducible, we opted for the Gaussian prior in this version. That said, we are open to new directions for layer selection.
>
> We added a discussion in Section 6 of the revision to highlight the above points.
>
> [1] What does BERT learn about the structure of language?
>
> [2] Transformer feed-forward layers are key-value memories
>
> ---
>
> Please note this response continues in the next comment.

---

> ### Author Response · Authors · 2025-11-20
> **Response to Reviewer 1XmS (Part 3)**
>
> ***Q4. Does the integration of KV pairs affect positional encoding or cause mismatch issues?***
>
> We thank the reviewer for bringing up this superbly detailed point. We completely agree that naïvely merging KV tensors may introduce positional misalignment. To address this, our implementation is already designed to preserve positional coherence across all layers.
>
> For the receiver model, in each layer, we shift all its positions by $|C|$, where $|C|$ is the length of the sender’s context. For selected layers, we concatenate the sender’s KV pairs at positions $[0,|C|)$, and the receiver’s KV pairs follow at positions $[|C|,|C|+|Q|)$. For non-selected layers, positions $[0,|C|)$ are left empty (unattended), but the receiver’s KV still starts at position $|C|$.
>
> This approach ensures that all layers share a consistent positional frame, so the attention mechanism sees the same offsets at every depth, avoiding positional drift across layers.
>
> To directly quantify the importance of this design, we performed an additional ablation experiment. For non-selected layers, instead of shifting the receiver’s tokens to position $|C|$, we place them back to position 0, creating a positional inconsistency with selected layers.
>
> The table below shows that positional inconsistency does not have a detrimental effect on performance, but overall, our approach has merit.
>
> (The table is formatted in a way that it first shows the source and receiver model pair, followed by results. There are multiple model pairs. KVComm-S shifts back token positions of non-selected layers to 0.)
> | **Method** | **Countries** | **Tipsheets** | **HotpotQA** | **QASPER** | **MuSiQuest** | **MultiField-QA-en** | **2WikiM-QA** | **TMATH** |
> |-----------|---------------|---------------|--------------|------------|----------------|-----------------------|----------------|-----------|
> **$\mathcal{M}_s$: huihui-ai/Llama-3.2-3B-Instruct-abliterated;**
> **$\mathcal{M}_r$: suayptalha/DeepSeek-R1-Distill-Llama-3B**
> | KVComm-S (0.3) | 0.26 | **0.65** | 0.40 | **0.10** | 0.14 | **0.19** | 0.21 | **0.36** |
> | KVComm (0.3)   | **0.46** | 0.45 | **0.46** | 0.09 | **0.28** | 0.15 | **0.28** | 0.35 |
> | KVComm-S (0.5) | 0.49 | 0.74 | **0.57** | **0.28** | **0.32** | 0.45 | 0.30 | **0.35** |
> | KVComm (0.5)   | **0.57** | **0.81** | **0.57** | 0.27 | **0.32** | **0.51** | **0.36** | **0.35** |
> | KVComm-S (0.7) | 0.52 | 0.76 | **0.65** | **0.30** | **0.39** | 0.46 | 0.32 | **0.35** |
> | KVComm (0.7)   | **0.57** | **0.81** | **0.65** | 0.29 | 0.36 | **0.47** | **0.37** | **0.35** |
> **$\mathcal{M}_s$: Orion-zhen/Qwen2.5-7B-Instruct-Uncensored;**
> **$\mathcal{M}_r$: bespokelabs/Bespoke-Stratos-7B**
> | KVComm-S (0.3) | 0.00 | 0.20 | **0.02** | **0.02** | **0.01** | **0.09** | 0.05 | **0.34** |
> | KVComm (0.3)   | **0.04** | **0.26** | **0.02** | 0.01 | 0.01 | **0.09** | **0.08** | 0.31 |
> | KVComm-S (0.5) | 0.04 | **0.90** | **0.33** | **0.13** | **0.18** | **0.35** | **0.16** | **0.35** |
> | KVComm (0.5)   | **0.19** | 0.88 | 0.28 | 0.07 | 0.12 | 0.26 | 0.10 | 0.33 |
> | KVComm-S (0.7) | 0.36 | **0.94** | **0.42** | 0.19 | 0.24 | **0.35** | **0.16** | **0.34** |
> | KVComm (0.7)   | **0.41** | 0.89 | 0.41 | **0.21** | **0.25** | 0.29 | 0.15 | **0.34** |
> **$\mathcal{M}_s$: ehristoforu/falcon3-ultraset;**
> **$\mathcal{M}_r$: huihui-ai/Falcon3-7B-Instruct-abliterated**
> | KVComm-S (0.3) | **0.47** | **0.71** | 0.54 | 0.10 | 0.36 | 0.19 | 0.26 | **0.32** |
> | KVComm (0.3)   | 0.46 | 0.69 | **0.59** | **0.19** | **0.40** | **0.35** | **0.29** | **0.32** |
> | KVComm-S (0.5) | 0.36 | **0.97** | **0.67** | **0.27** | **0.46** | 0.34 | **0.36** | 0.34 |
> | KVComm (0.5)   | **0.40** | 0.92 | 0.63 | 0.25 | 0.44 | **0.45** | 0.34 | **0.35** |
> | KVComm-S (0.7) | **0.21** | 0.95 | **0.59** | **0.26** | **0.52** | 0.46 | **0.37** | **0.36** |
> | KVComm (0.7)   | 0.19 | **0.96** | 0.55 | **0.26** | 0.42 | **0.51** | 0.31 | **0.36** |
>
> We included this clarification in Appendix K and the new ablation results in Appendix M of the revised manuscript uploaded on 20 Nov.
>
> ---
>
> Please note this response continues in the next comment.

---

> ### Author Response · Authors · 2025-11-20
> **Response to Reviewer 1XmS (Part 4)**
>
> ***Q5. What is the impact on latency and memory footprint during runtime?***
>
> Thank you for raising this issue, which is important for real deployment. In the paper (Figure 8), we have already reported that KVComm achieves 2.5x to 6x FLOPs reduction compared to the Skyline. A similar advantage also holds for KV memory usage. KVComm only transmits and stores a selected subset of KV layers on the receiver side.
>
> This means that the KV memory footprint grows proportionally to the selected ratio rather than the full model depth. KVComm does not require re-prefill, which keeps memory usage minimal during inference.
>
> We added a new analysis for memory to **Figure 8** of the revised manuscript uploaded on 20 Nov. As shown in the figure, KVComm also has a significant memory advantage over Skyline, with 23% to 73% less memory usage than Skyline, especially when selecting fewer layers for communication.
>
> Admittedly, KVComm has a larger memory footprint than the absolute baseline that only processes the query, but such an approach clearly suffers performance-wise in terms of expressiveness of communication.
>
> We changed Figure 8 and clarified this point in Section 4.6 of the revised manuscript uploaded on 20 Nov.

---

> > ### Comment · Reviewer_1XmS · 2025-11-28
> > **ack**
> >
> > I thank the authors for their effective rebuttal. I have no more questions. thanks.

---

### Official Review · Reviewer_1arg · 2025-11-01

**Soundness:** 3
**Presentation:** 3
**Contribution:** 2
**Rating:** 6
**Confidence:** 3

**Summary:**

In this paper, the author studies the interesting problem of inefficient communication in multi-agent LLM systems. The authors argue that existing methods, such as using natural language communication, suffer from high inference costs and information loss , while methods using hidden states of last token can be ineffective. To address this, the authors propose KVComm, a framework for communication via the selective sharing of KV pairs from a sender model to a receiver model. The framework is based on identifying the most informative layer's KV pairs to transmit. This strategy assumes that intermediate layers contain the most transferable semantic knowledge, and that layers with higher attention paid to the context are more valuable. Based on evaluation on 6 benchmarks, the paper highlights the efficacy of the proposed approach.

**Strengths:**

1. The paper is well-written in general and the research problem has been well-articulated to the reader. It addresses an important and timely topic, communication among large language model (LLM) agents, which is relevant to the community given the growing interest in multi-agent systems.

2. The proposed framework of using KV pairs as the medium for communication is intuitive and also looks technically sound.

3. The paper shows dominant performance over existing communication protocols like AC, NLD, and CIPHER across a diverse set of tasks and model pairs demonstrating its efficacy. Further, the authors perform several ablations to support the claims.

**Weaknesses:**

1. The only concern is whether the proposed method generalizes to scenarios where the sender and receiver are LLMs with different architectures. In the experiments, the authors state that they "limit the choices of the two LLMs to (1) two instances of the same LLM, and (2) two models that are fine-tuned versions of the same base LLM". If its a limitation then the authors should clearly mention it.

2. The KV cache sharing framework has some conceptual similarities with a prior work [1]. The authors are encouraged to clarify the distinctions of their approach relative to that work to better highlight the novelty.


[1] DroidSpeak: KV Cache Sharing for Cross-LLM Communication and Multi-LLM Serving

**Questions:**

Please clarify weakness 1.

---

> ### Author Response · Authors · 2025-11-20
> **Response to Reviewer 1arg**
>
> Thank you for your time and your thoughtful review. Below, we provide detailed responses to each of your comments.
>
> ---
>
> ***Q1. The only concern is whether the proposed method generalizes to scenarios where the sender and receiver are LLMs with different architectures. In the experiments, the authors state that they "limit the choices of the two LLMs to (1) two instances of the same LLM, and (2) two models that are fine-tuned versions of the same base LLM". If its a limitation then the authors should clearly mention it.***
>
> Thank you for raising this important question. We confirm this as a limitation. As part of our preliminary work, we did consider the case of communication across LLMs with embedding spaces. Our preliminary results were positive, but we decided to break down the problem in such a way that we first demonstrate the possibility of efficiently and effectively communicating across models. Having evaluated the effectiveness of transmitting hidden states, we focused on KV as the key source of information. Thus, we initially chose models that belong to the same architecture, because KV pair structures differ significantly across architectures. We leave the problem of cross-architecture KV alignment to future work, which may leverage latent space projection or learnt adapters. To address this comment, we explicitly added a paragraph in Section 6 of the revised manuscript.
>
> ---
>
> ***Q2. The KV cache sharing framework has some conceptual similarities with a prior work [1]. The authors are encouraged to clarify the distinctions of their approach relative to that work to better highlight the novelty.***
>
> We thank the reviewer for asking us to further differentiate our work from DroidSpeak [1]. Succinctly, KVComm enables non-contiguous KV cache layer selection (as opposed to DroidSpeak’s single contiguous chunk), and we demonstrate the distinct benefit of our approach. Below is a detailed clarification of the differences:
> * Mechanism:
>     * DroidSpeak empirically selects only a single contiguous chunk of layers’ KV pairs for reuse, while recomputing the KV for the rest of the layers. In addition, DroidSpeak has a large calibration overhead.
>     * KVComm flexibly selects KV pairs of **non-contiguous layers** based on attention importance and a Gaussian prior with low overhead, without needing to recompute the remaining layers. Our method can be effectively calibrated with a single sample.
> * Objective & Problem Setting:
>     * DroidSpeak focuses on accelerating inference for queries with shared prefixes. It shares the partial KV cache of these prefixes among different queries to avoid redundant recomputation.
>     * We focus on communication between agents and identify which intermediate layer KV pairs carry maximal transferable information for multi-agent cooperation. We selectively share the KV pairs of the context between agents for contextual task solving.
>
> **Experimental Comparison with DroidSpeak**: Notably, in Figure 5, we show that **KVComm can achieve the Pareto optimal frontier compared to all combinations of single contiguous chunks**, which serves as an approximation upper-bound of what DroidSpeak does. This directly demonstrates the advantage of our strategy.
>
> We highlight our novelty and contribution as follows:
> * We evaluate different types of activation information for communication between LLMs, and identify the limitations of using hidden states as the medium of communication, which serves as the motivation for using KV pairs.
> * We propose selective non-contiguous KV sharing guided by attention importance scores and a Gaussian prior. A single context/question pair can have generalizable calibration.
> * We demonstrate that KVComm can match or exceed the performance of upper-bound Skyline, while reducing the computation costs by 2.5x to 6x.
>
> We added a clearer discussion in Section 5 related work to avoid any ambiguity.

---

> ### Author Response · Authors · 2025-11-27
> **Executive summary of our response and seeking additional feedback**
>
> Dear reviewer,
>
> We hope you are doing well. To very briefly summarize our response: Our work enables non-contiguous KV cache layer selection (as opposed to DroidSpeak’s single contiguous chunk), and we demonstrate the distinct benefit of our approach. In particular, in Figure 5, we show that KVComm can achieve the Pareto optimal frontier compared to all combinations of single contiguous chunks, which serves as an approximation upper-bound of what DroidSpeak does. We have also clarified the limitation of not supporting heterogeneous model architectures.
>
> We are very happy to hear if you have any additional feedback.

---

> ### Comment · Reviewer_1arg · 2025-11-28
> **Official Comment by Reviewer 1arg**
>
> Thank you for your response. The rebuttal clears all my concerns. I shall increase my score accordingly.

---

### Author Response · Authors · 2025-11-21
**Response Summary**

We sincerely thank all reviewers for their comments and especially suggestions for improving our paper. Besides addressing each comment in per-review comment(s), we have updated the paper and clearly marked our changes in blue. In the updated PDF we have also indicated which review comment is being addressed by each change, and we hope that, combined with our comments in the system, this way of marking the changes will make it much easier for reviewers to check how we have addressed their feedback. This is especially the case for new figures that we could not inline in the OpenReview system.

Multiple reviewers questioned the ability of our approach to work in a multi-agent setting (as opposed to having just one sending and one receiving model as we already demonstrated). We have taken this opportunity to demonstrate the ease of adapting our approach to having two senders communicate their selected KV pairs to one receiver. The resulting 2-to-1 performance typically exceeds that of the 1-to-1 approach we have previously shown.

Also, multiple reviewers wanted us to clarify whether our approach currently works across models that are heterogeneous (i.e., come from different baselines). We acknowledge that our current prototype does not have this functionality, and we leave adapting it to work in the heterogeneous setting as future work.

Our responses and clarifications are numerous, and besides the 2-to-1 case, include evaluation on an additional model, introduction of expanded datasets, quantification of the memory consumption, explanation of prior choices, exploration of the possibility of dynamic layer selection, clarification/evaluation of the way our approach carefully integrates KV pairs while adjusting the positional encoding, and revised Natural Language Debate (NLD) performance that is still inferior to our approach.

Finally, we explained how our work differs from the references that the reviewers have thoughtfully brought up. For example, our work decisively goes beyond DroidSpeak [1] and the contemporaneous work similarly titled KVCOMM [2] (published very recently at NeurIPS 2025) by making it possible to efficiently and selectively choose layers of KV pairs that will be transmitted (as opposed to transmitting the entire KV or a contiguous chunk of it).

[1] DroidSpeak: KV Cache Sharing for Cross-LLM Communication and Multi-LLM Serving

[2] KVCOMM: Online Cross-context KV-cache Communication for Efficient LLM-based Multi-agent Systems

---

### Author Response · Authors · 2025-12-03
**Final Summary for the Area Chair**

Dear AC,

To help you with the difficult task of going through all the reviews and our responses, we provide the following summary. Please note that reviewers `1XmS`, `1arg`, and `XrXt` were satisfied with our responses (and the latter 2 wanted to raise their scores), while reviewer `i79Z` did not respond.

# Recognized Strengths

- Reviewers found Communication with KV pairs to be a **promising, compelling, and less-explored direction**. (`1arg`, `1XmS`, `i79Z`, `XrXt`)

- Reviewers found the layer selection strategy based on attention importance scores with a Gaussian prior was **well motivated and empirically validated**. Preliminary analyses, such as token-importance studies and the limitations of hidden state communication, further justify the design choices. (`1arg`, `1XmS`, `i79Z`)

- Our paper demonstrates **robust performance advantages** of our approach when extensively compared to strong protocols (e.g., Skyline, AC, NLD, CIPHER), and includes supporting ablations. In particular, KVComm achieves consistent, measurable **communication** (**up to 3x reduction**) and **computation** (**2.5x to 6x reduction**) gains over prior baselines, with broad evaluations across multiple datasets and model pairs. (`1arg`, `1XmS`, `i79Z`, `XrXt`)

# Response to Reviewers’ Main Concerns

- **Multiple agent setting** (`XrXt` Q1)

We clarify that KVComm can easily **extend to multiple senders and multiple receivers**. We demonstrate this by extending our approach to the **two-to-one** setting, where two senders jointly calibrate with one receiver, and then transmit selected KV pairs to the receiver. The results, added in **Appendix J**, show that KVComm maintains its effectiveness in this multi-agent scenario. Moreover, we found that two-to-one outperforms one-to-one settings for 17 out of 27 cases.


- **More datasets and model pairs** (`1XmS` S3, `XrXt` Q6)

At reviewers' request, we have added results on three expanded datasets (**HotpotQA-E, QASPER-E, and MuSiQuest-E**), a summarization dataset **SAMSum**, and the official **DeepSeek-R1-Distill-Llama-8B** model. The results are consistent with our prior findings, showing KVComm's effectiveness across diverse settings. We incorporated these results in **Appendix E and F**.

- **Communication among heterogeneous model architectures** (`1arg` Q1, `1XmS` S1&Q1, `i79Z` Q4, `XrXt` Q1)

We acknowledge that our current prototype limits the choice of models to those finetuned from the same base model. We leave the problem of heterogeneous architecture KV alignment to future work, which may leverage latent space projection or learnt adapters. We explicitly added a discussion in Section 6.

- **Latency and memory footprint** (`1XmS` Q5)

We added detailed analyses for both **latency and memory footprint** in **Figure 8**. The results show that KVComm achieves **2.5x to 6x FLOPs reduction**, and **23% to 73%** memory reduction over Skyline, leading to significant latency improvements.

- **NLD baseline** (`i79Z` Q3)

Upon reviewer request, we revised our NLD baseline with correct implementation and information-transfer style discussion to ensure a **fair comparison**. We found that NLD cannot reliably extract and transmit the task-relevant subset of information, but can only perform competently when the required amount of information is small. We additionally provide ablation results in **Table 1** and **Figure 12** on the effect of **communication length** for NLD to further validate our findings.

- **Comparison to related works** (`1arg` Q2, `i79Z` Q1, `XrXt` Q7)

`1arg` mentioned DroidSpeak[1] in their Q2. Succinctly, KVComm enables **non-contiguous KV cache layer selection** (as opposed to DroidSpeak’s **single contiguous chunk**), and we demonstrate the distinct benefit of our approach in **Figure 5** of our manuscript.

At the request of `i79Z`, we clarify the differences between KVComm and single-model sparse attention or cache reuse by highlighting that KVComm focuses on **different computational settings** and **distinct objectives**, with the possibility of **cross-model communication**.

`XrXt` mentioned [2] in their Q7. Our work goes beyond [2] by: 1) enabling a **different type of communication** (where the receiver does not have the context), 2) making it possible to efficiently and selectively choose **layers of KV pairs that will be transmitted**, and 3) being able to work effectively across **different fine-tuned** models.

We added a discussion in Section 5 for clarification.

[1] DroidSpeak: KV Cache Sharing for Cross-LLM Communication and Multi-LLM Serving

[2] KVCOMM: Online Cross-context KV-cache Communication for Efficient LLM-based Multi-agent Systems

- **Additional Comments**

Besides the points above, our responses include **expansion to online calibration** (`1XmS` S2&Q2), **alignment of positional encodings** (`1XmS` Q3), explanation of Gaussian prior choices (`1XmS` Q4), and **the choice of KV transmission over natural language** (`i79Z` Q1).

---

### Meta-Review · Area_Chair_RANz · 2026-01-03

**Summary:**

This paper introduces KVComm, a selective KV-cache sharing framework for efficient inter-LLM communication.

Reviewers consistently found the problem important, the approach intuitive yet technically well-motivated, and the empirical evaluation broad and compelling. KVComm achieves substantial reductions in communication and computation cost while maintaining accuracy close to the upper-bound “Skyline” method, outperforming a wide range of strong baselines (AC, NLD, CIPHER). The authors conducted extensive additional experiments, including multi-agent (2-to-1) communication, expanded datasets, official model releases, and latency/memory analyses, which reinforce the strength and generality of the framework. Overall, the submission is well-executed, demonstrates clear novelty within the KV-sharing design, and shows strong practical relevance.

**Reviewer Concerns:**

The authors thoroughly addressed nearly all substantive concerns. Limitations regarding heterogeneous architectures were acknowledged explicitly, with a clear plan for future extensions. The multi-agent concern was resolved through new experiments showing effective 2-to-1 communication, and several reviewers remarked that this fully addressed their reservations. Clarifications on NLD, attention-importance formalization, positional encoding handling, and KV integration were detailed and sufficient. Additional experiments like expanded datasets, official DeepSeek models, multi-agent calibration, and dynamic selection analysis further strengthened the empirical picture. The one reviewer who could not update the score explicitly stated they would communicate a more positive assessment.

**Reviewer Scores:**

Reviewer 1arg: Raised score (from 6 to higher; strong accept tendency).

Reviewer 1XmS: Remains strongly positive (8).

Reviewer i79Z: Likely unchanged at (4), but concerns largely addressed, the tone is neutral-to-positive.

Reviewer XrXt: Initially (4), could not change score due to system restrictions, but explicitly stated they would raise their rating (maybe 4 -> 6).

---

### Decision · Program_Chairs · 2026-01-26

Accept (Poster)